# DRB1, DRB2 and DRB4 Are Required for Appropriate Regulation of the microRNA399/*PHOSPHATE2* Expression Module in *Arabidopsis thaliana*

**DOI:** 10.3390/plants8050124

**Published:** 2019-05-13

**Authors:** Joseph L. Pegler, Jackson M. J. Oultram, Christopher P. L. Grof, Andrew L. Eamens

**Affiliations:** Centre for Plant Science, School of Environmental and Life Sciences, Faculty of Science, University of Newcastle, Callaghan 2308, New South Wales, Australia; Joseph.Pegler@uon.edu.au (J.L.P.); Jackson.Oultram@uon.edu.au (J.M.J.O.); chris.grof@newcastle.edu.au (C.P.L.G.)

**Keywords:** *Arabidopsis thaliana*, phosphorous (P), phosphate (PO_4_) stress, microRNA (miRNA), miR399, *PHOSPHATE2* (*PHO2*), DOUBLE-STRANDED RNA BINDING (DRB) proteins DRB1, DRB2, DRB4, miR399-directed *PHO2* expression regulation, RT-qPCR

## Abstract

Adequate phosphorous (P) is essential to plant cells to ensure normal plant growth and development. Therefore, plants employ elegant mechanisms to regulate P abundance across their developmentally distinct tissues. One such mechanism is PHOSPHATE2 (PHO2)-directed ubiquitin-mediated degradation of a cohort of phosphate (PO_4_) transporters. *PHO2* is itself under tight regulation by the PO_4_ responsive microRNA (miRNA), miR399. The DOUBLE-STRANDED RNA BINDING (DRB) proteins, DRB1, DRB2 and DRB4, have each been assigned a specific functional role in the *Arabidopsis*
*thaliana* (*Arabidopsis*) miRNA pathway. Here, we assessed the requirement of DRB1, DRB2 and DRB4 to regulate the miR399/*PHO2* expression module under PO_4_ starvations conditions. Via the phenotypic and molecular assessment of the knockout mutant plant lines, *drb1*, *drb2* and *drb4*, we show here that; (1) DRB1 and DRB2 are required to maintain P homeostasis in *Arabidopsis* shoot and root tissues; (2) DRB1 is the primary DRB required for miR399 production; (3) DRB2 and DRB4 play secondary roles in regulating miR399 production, and; (4) miR399 appears to direct expression regulation of the *PHO2* transcript via both an mRNA cleavage and translational repression mode of RNA silencing. Together, the hierarchical contribution of DRB1, DRB2 and DRB4 demonstrated here to be required for the appropriate regulation of the miR399/*PHO2* expression module identifies the extreme importance of P homeostasis maintenance in *Arabidopsis* to ensure that numerous vital cellular processes are maintained across *Arabidopsis* tissues under a changing cellular environment.

## 1. Introduction

Phosphorous (P) is one of the most limiting factors for plant growth worldwide [1,2,3], with large quantities of P an essential requirement for numerous processes vital to the plant cell, including energy trafficking, signaling cascades, enzymatic reactions and nucleic acid and phospholipid synthesis [3,4]. Inorganic phosphate (Pi), in the form of PO_4_, is the predominant form of P taken up by a plant from the soil, however, soil PO_4_ primarily exists in organic or insoluble forms that are largely inaccessible to plant root uptake mechanisms [1]. Therefore, due to limited soil PO_4_ availability, combined with the importance of an adequate concentration of P in plant cells to ensure normal growth and development, plants employ elegant mechanisms to spatially regulate P abundance across their developmentally distinct tissues [5,6]. Phosphorous homeostasis is therefore tightly controlled and involves both the remobilization of internal P stores and the increased acquisition of external PO_4_ [5,7]. For example, P limitation triggers the release of organic acids from the plant root system into the soil rhizosphere to chelate with metal ions to promote soluble PO_4_ uptake to maintain or increase intracellular P concentration [1,8]. In addition, the P stored in the older leaves of a plant when the plant experiences P stress is remobilized; this allows for (1) continued growth of actively expanding tissues, and (2) the promotion of new growth. Enhanced P trafficking is achieved via promoting the expression of genes encoding PO_4_ transporter proteins, and in turn, elevated PO_4_ transporter protein abundance generally ensures that the cellular P concentration is maintained irrespective of external PO_4_ levels [1,7].

In *Arabidopsis thaliana* (*Arabidopsis*), the first protein identified to be required for the maintenance of P homeostasis under PO_4_ limiting conditions was PHOSPHATE1 (PHO1) [9]. The gene encoding PHO1 (*PHO1*; *AT1G14040*) was identified by [9] via their characterization of *pho1* plants, an *Arabidopsis* mutant line demonstrated to over-accumulate P in root tissues due to defective P translocation to the shoot. Although the *Arabidopsis* PHO1 protein, and the PHO1 proteins of other plant species characterized to date, do not closely resemble other PO_4_ transporter proteins, PHO1 is indeed central to P movement in plants. The PHO1 protein is essential for PO_4_ efflux into the root vascular cylinder; the first step in P transportation to the upper aerial tissues [10,11]. PHOSPHATE2 (PHO2) was the second protein demonstrated essential for the maintenance of P homeostasis with the *pho2* mutant shown to accumulate P to toxic levels in shoot tissues [12,13]. The *PHO2* gene (*AT2G33770*) has since been shown to encode a ubiquitin conjugating enzyme24 (UBC24), with the PHO2 UBC24 proposed to direct ubiquitin-mediated degradation of PO_4_ transporters, PHOSPHATE TRANSPORTER1;4 (PHT1;4), PHT1;8 and PHT1;9 [14]. Further, *PHO2* is almost ubiquitously expressed in *Arabidopsis* shoot and root tissues [15], with the loss of PHO2-directed suppression of PHT1;4, PHT1;8 and PHT1;9 abundance in *pho2* plants leading to the enhanced translocation of P from the roots to the shoot tissue [14]. In addition to PHO1 and PHO2, traditional mutagenesis-based approaches have further identified other proteins essential to P homeostasis maintenance, including PHOSPHATE STARVATION RESPONSE1 (PHR1), a MYB domain transcription factor that regulates the expression of numerous P responsive genes [16,17].

More contemporary research, however, has concentrated on documenting the regulatory role directed at the posttranscriptional level by small regulatory RNAs (sRNA), specifically the microRNA (miRNA) class of sRNA, in order to maintain P homeostasis [18,19]. The advent of high throughput sequencing technologies has made sRNA profiling across plant species, and under different growth regimes, including exposure of a plant to abiotic and biotic stress, a routine experimental procedure in modern research [14,20,21]. Such profiling has identified a common suite of conserved miRNAs (miRNAs identified across multiple, evolutionary unrelated plant species) that accumulate differentially when mineral nutrients are lacking, including P, nitrogen (N), copper and sulphur [20,21]. Responsiveness of a single miRNA to multiple mineral nutrient stresses is not surprising considering the considerable overlap in the complex regulation of metal ion transport and/or uptake in plants [14,22,23]. In *Arabidopsis* for example, P and N uptake mechanisms are reciprocally linked to one another, therefore; a miRNA with enhanced accumulation during periods of P stress will usually be reduced in abundance during N starvation [19,24,25].

The miRNA, miR399, has been conclusively linked with the maintenance of P homeostasis and the regulation of PO_4_ uptake in *Arabidopsis* [18,19]. In *Arabidopsis*, the miR399 sRNA is processed from six precursor transcripts, namely *PRE-MIR399A* to *PRE-MIR399F*, transcribed from five genomic loci (*MIR399A*-*MIR399D* and *MIR399E/F*). The miR399 sRNA is unique amongst *Arabidopsis* miRNAs in that it acts as a mobile systemic signal upon PO_4_ stress [21,26]. More specifically, when P becomes limited in *Arabidopsis* shoots, *MIR399* gene expression is stimulated by PHR1 [27], and following processing of the now abundant miR399 precursor transcripts by the protein machinery of the *Arabidopsis* miRNA pathway, the mature miR399 sRNA is transported to the roots. Here, miR399 is actively loaded by the miRNA-induced silencing complex (miRISC) to direct miRISC-mediated cleavage of *PHO2*, the target transcript of miR399 [7,21,27]. Reduced PHO2 protein abundance, due to elevated miRISC-mediated cleavage of the *PHO2* transcript, in turn removes the PHO2-mediated suppression of PO_4_ transporters, PHT1;4, PHT1;8 and PHT1;9, to ultimately promote root-to-shoot P transport in an attempt to maintain shoot P homeostasis in P limited conditions [28,29,30,31]. Additional regulatory complexity to the miR399/*PHO2* expression module is offered by the non-protein-coding RNA, *INDUCED BY PHOSPHATE STARVATION1* (*IPS1*) [32]. Once transcribed, *IPS1* acts as an endogenous target mimic (eTM) of miR399 activity [33]. Specifically, the miR399 target site harbored by *IPS1* contains a three nucleotide mismatch bulge across miR399 nucleotide positions 10 and 11: the position at which the catalytic core of miRISC, ARGONAUTE1 (AGO1), catalyzes the cleavage of miRNA target transcripts [34]. The bulge that forms at this position once miR399-directed AGO1 binds *IPS1*, renders *IPS1* resistant to AGO1-catalyzed cleavage, thereby effectively sequestering away miR399 activity [33].

Three of the five members of the *Arabidopsis* DOUBLE-STRANDED RNA BINDING (DRB) protein family, including DRB1, DRB2 and DRB4, have been assigned functional roles in the *Arabidopsis* miRNA pathway [35,36,37,38,39]. Both DRB1 and DRB4 form functional partnerships with DICER-LIKE (DCL) proteins, RNase III-like endonucleases that cleave molecules of double-stranded RNA (dsRNA). More specifically, the DRB1/DCL1 partnership processes stem-loop structured molecules of imperfectly dsRNA that form post miRNA precursor transcript folding [35,36,37], and the DRB4/DCL4 partnership is central for the processing of a small subset of miRNA precursor transcripts that fold to form stem-loop structures with high levels of base-pairing due to the almost perfect complementarity of the nucleotide sequences of the stem-loop arms [39]. More recently, DRB2 has also been assigned a functional role in the *Arabidopsis* miRNA pathway due to its demonstrated antagonism and/or synergism with the roles of both DRB1 and DRB4 in sRNA production [37,40]. Here, we therefore assessed the requirement of DRB1, DRB2 and DRB4 in the regulation of the miR399/*PHO2* expression module, both under non-stressed growth conditions and when wild-type *Arabidopsis* plants (ecotype Columbia-0 (Col-0)) and the *drb1*, *drb2* and *drb4* mutant lines are exposed to PO_4_ starvation. More specifically, we aimed to determine; (1) the contribution of DRB1, DRB2 and/or DRB4 to miR399 production; (2) the mode of silencing directed by miR399 to regulate *PHO2* expression, and; (3) whether either DRB1, DRB2 or DRB4 are required for P homeostasis maintenance. Phenotypic and molecular assessment of Col-0, *drb1*, *drb2* and *drb4* plants post exposure to a 7-day period of PO_4_ starvation, revealed that DRB1 and DRB2 are required for P homeostasis maintenance. Further, DRB1 was established as the primary DRB protein required to regulate miR399 production. However, DRB2 and DRB4 were demonstrated to play a secondary role in miR399 production regulation. Furthermore, miR399 appears to regulate the expression of its targeted transcript, *PHO2*, via both the canonical mechanism of plant miRNA-directed target gene expression repression, target mRNA cleavage, and via the alternative mode of target gene expression regulation, translational repression. Taken together, the hierarchical contribution of DRB1, DRB2 and DRB4 to the regulation of the miR399/*PHO2* expression module in *Arabidopsis* shoots and roots identifies the extreme importance of maintaining P homeostasis to ensure that numerous vital cellular processes are maintained across *Arabidopsis* tissue types and under a changing cellular environment.

## 2. Results

### 2.1. The Phenotypic and Physiological Response to PO_4_ Stress in the Shoot Tissues of Arabidopsis Plant Lines Defective in DRB Protein Activity

To determine the consequence of loss of DRB activity on P homeostasis maintenance in 15-day old *Arabidopsis* plants post a 7-day period of PO_4_ starvation, a series of phenotypic and physiological parameters were assessed in Col-0, *drb1*, *drb2* and *drb4* shoots. The severe developmental phenotype of the *drb1* mutant has been reported previously [36,41,42]. Figure 1A clearly reveals the reduced size of the *drb1* mutant at 15 days of age, compared to Col-0 plants, when both *Arabidopsis* lines are cultivated on standard growth media (P^+^ media). The retarded development of the *drb1* mutant is further evidenced in Figure 1B where the fresh weight of 8-day old Col-0 and *drb1* seedlings is presented. Specifically, prior to seedling transfer to either P^+^ or P^−^ media, the fresh weight of an 8-day old *drb1* seedling (13.5 ± 1.0 mg) is 53.4% less than that a Col-0 seedling (29.0 ± 3.5 mg). Compared to *drb1*, the *drb2* and *drb4* mutants display mild developmental phenotypes [37,42] as evidenced by those displayed by 15-day old *drb2* and *drb4* plants cultivated on P^+^ growth media (Figure 1A), and by the fresh weights of 8-day old *drb2* (26.8 ± 4.2 mg) and *drb4* (22.9 ± 1.4 mg) seedlings. Although the *drb1* mutant displayed the most severe phenotype, *drb1* development appeared to be the least affected by the 7-day PO_4_ stress treatment. The fresh weight of P^−^
*drb1* plants (35.5 ± 1.0 mg) was only reduced by 21.6% compared to P^+^
*drb1* plants (45.3 mg ± 1.5 mg) (Figure 1C). The development of Col-0, *drb2* and *drb4* plants was negatively impacted to a similar degree by the 7-day PO_4_ stress treatment, with their fresh weights reduced by 36.6%, 39.1% and 36.3%, respectively (Figure 1C). Determination of rosette area revealed largely similar trends across the *drb* mutant lines analyzed, that is, *drb1* rosette area was reduced by 29.3%, while the rosette development of P^−^
*drb2* and P^−^
*drb4* plants was reduced by 48.0% and 38.7%, respectively (Figure 1D). Interestingly, the observed reductions to the rosette area of P^−^
*drb1*, P^−^
*drb2* and P^−^
*drb4* plants was considerably less than the 60.1% reduction to the rosette area of P^−^ Col-0 plants (11.2 ± 1.7 mm^2^) compared to P^+^ Col-0 plants (28.1 ± 5.5 mm^2^) (Figure 1D).

Anthocyanin, chlorophyll *a* and chlorophyll *b* content of Col-0, *drb1*, *drb2* and *drb4* shoots was also determined. Phosphate starvation has been previously shown to elevate the levels of PRODUCTION OF ANTHOCYANIN PIGMENT1 (PAP1/MYB75), PAP2 (MYB90) and MYB113, three MYB domain transcription factors that in turn stimulate the expression of a cohort of genes required for anthocyanin production in vegetative tissues [19,43]. These reports, in combination with the readily observable pigmentation that accumulated in the rosette leaves of P^−^ Col-0, P^−^
*drb2* and P^−^
*drb4* plants (Figure 1A), identified anthocyanin as an ideal metric to further assess the response of each *drb* mutant to PO_4_ starvation. The anthocyanin content of non-stressed Col-0, *drb1*, *drb2* and *drb4* shoots was similar (Figure 1E). However, when PO_4_ is limited, an approximate 2.0-fold increase in anthocyanin accumulation was detected for P^−^ Col-0 shoots. Further promotion of anthocyanin accumulation was determined for PO_4_-stressed *drb2* and *drb4* plants, with anthocyanin content elevated 3.7- and 2.8-fold in P^−^
*drb2* and P^−^
*drb4* plants, respectively (Figure 1E). As readily observable in Figure 1A, anthocyanin accumulation was not promoted in the shoot tissue of P^−^
*drb1* plants. However, spectrophotometry revealed abundance changes for both chlorophyll *a* and chlorophyll *b* in the shoot tissue of P^−^
*drb1* plants. Specifically, chlorophyll *a* (Figure 1F) and chlorophyll *b* (Figure 1G) abundance was elevated by 2.1- and 2.8-fold in P^−^
*drb1* shoots, compared to P^+^
*drb1* shoots. In PO_4_-stressed Col-0, *drb2* and *drb4* shoots, the chlorophyll *a* level remained largely unchanged compared to the non-stressed counterpart of each plant line (Figure 1F). Chlorophyll *b* accumulation however, was determined to be promoted in Col-0 and *drb4* shoots, by 1.8- and 2.0-fold, by the 7-day PO_4_ starvation period (Figure 1G).

### 2.2. Molecular Profiling of the miR399/PHO2 Expression Module in the Shoot Tissues of Arabidopsis Plant Lines Defective in DRB Protein Activity

The results presented in Figure 1 strongly indicated that each *drb* mutant was responding differently to the applied stress and when this finding is considered together with the documented roles of DRB1, DRB2 and DRB4 in the *Arabidopsis* miRNA pathway [35,36,37,38,39], including the demonstrated antagonism between DRB1 and DRB2 [37] and between DRB2 and DRB4 [40] in miRNA production, the miR399/*PHO2* expression module was next profiled via a RT-qPCR-based approach. RT-qPCR profiling was conducted in an attempt to determine if the observed differences in the response of each *drb* mutant line to PO_4_ stress was a result of dysfunction of the miR399/*PHO2* expression module.

In *Arabidopsis* shoots, *PHR1* promotes *MIR399* gene expression when PO_4_ supplies become limited, resulting in elevated miR399 abundance [27]. Therefore, RT-qPCR was first used to assess *PHR1* expression in control and PO_4_-stressed Col-0, *drb1*, *drb2* and *drb4* shoots (Figure 2A). *PHR1* expression was only mildly elevated by 1.5-, 1.6- and 1.7-fold in P^+^
*drb1*, P^+^
*drb2* and P^+^
*drb4* shoots respectively, compared to its levels in non-stressed Col-0 shoots (Figure 2A). RT-qPCR further revealed that PO_4_ stress only induced mild elevations to *PHR1* expression in P^−^ Col-0 (1.00 to 1.22 relative expression) and P^−^
*drb2* shoots (1.62 to 1.74 relative expression) (Figure 2A). This result was not unexpected in view of the previous report of only mild *PHR1* expression induction in PO_4_-stressed *Arabidopsis* [17]. Interestingly, *PHR1* expression was reduced by 19.6% and 31.2% in P^−^
*drb1* and P^−^
*drb4* shoots, respectively (Figure 2A), and not mildly elevated as expected.

The miR399 sRNA is processed from six structurally distinct precursor transcripts (*PRE-MIR399A* to *PRE-MIR399F*), transcribed from five genomic loci (*MIR399A* to *MIR399D* and *MIR399E/F*) in *Arabidopsis*. RT-qPCR only failed to detect *PRE-MIR399B* expression in Col-0 shoots. RT-qPCR did however clearly reveal that PO_4_ stress induced the expression of the five detectable miR399 precursor transcripts by 4.0-, 88.3-, 3204-, 37.3- and 92.9-fold in the shoots of P^−^ Col-0 plants (Figure 2B–F). Of the three members of the *Arabidopsis* DRB protein family analyzed here, Figure 2B–F clearly show that DRB1 is the primary DRB protein required to regulate miR399 production in *Arabidopsis* shoots with the abundance of *PRE-MIR399A*, *PRE-MIR399C*, *PRE-MIR399D*, *PRE-MIR399E* and *PRE-MIR399F* elevated by 2.3-, 10.1-, 12.8-, 5.5- and 14.6-fold, respectively, in P^+^
*drb1* shoots. The primary role of DRB1 in regulating miR399 production in *Arabidopsis* shoots was further highlighted for *PRE-MIR399A*, *PRE-MIR399C*, *PRE-MIR399D* and *PRE-MIR399F* via additional elevations to their respective expression levels, specifically 45.7-, 234.6- 3743- and 178.9-fold increases to transcript abundance in P- *drb1* shoots (Figure 2B–D,F).

Failure to detect the *PRE-MIR399A* precursor by RT-qPCR in P^+^
*drb2* shoots, and a similar degree of over-accumulation of this precursor in P^−^ Col-0 (4.0-fold) and P^−^
*drb2* shoots (4.6-fold), indicated that DRB2 is not required to regulate miR399 production from this precursor (Figure 2B). Wild-type-like accumulation of *PRE-MIR399C* (1.1-fold) and *PRE-MIR399D* (1.2-fold) in P^+^
*drb2* shoots, and a lower degree of over-accumulation of these two precursors in P^−^
*drb2* shoots, compared to P^−^ Col-0 shoots, indicated that DRB2 plays a secondary role in regulating miR399 production from these two precursors (Figure 2C,D). A similar level of expression of *PRE-MIE399E* in PO_4_-stressed *drb1* and *drb2* shoots suggested that both DRB1 and DRB2 are required for miR399 production from this precursor (Figure 2E). However, lower transcript abundance (0.5 relative expression) in P^+^
*drb2* shoots, compared to relative expression levels of 1.0 and 5.5 in P^+^ Col-0 and P^+^
*drb1* shoots, respectively (Figure 2E), again indicated that under standard growth conditions, DRB2 plays a secondary role in regulating miR399 production from the *PRE-MIR399E* precursor. The abundance of the *PRE-MIR399F* transcript is also reduced in P^+^
*drb2* shoots compared to its levels in P^+^ Col-0 shoots, and further, the degree of over-accumulation of *PRE-MIR399F* is less in P^−^
*drb2* shoots compared to its levels in P^−^ Col-0 shoots (Figure 2F). When these expression trends are considered together with those documented for P^+^ and P^−^
*drb1* shoots, they again indicate a secondary role for DRB2 in regulating miR399 production from this precursor.

As demonstrated for P^+^
*drb2* shoots, the *PRE-MIR399A* transcript remained below the detection sensitivity of RT-qPCR in P^+^
*drb4* shoots (Figure 2B). RT-qPCR did however, reveal *PRE-MIR399A* expression to be elevated by 5.2-fold in P^−^
*drb4* shoots, a similar degree of transcript elevation to that observed in P^−^ Col-0 shoots (4.0-fold increase) (Figure 2B). This indicates that DRB4 is not involved in regulating miR399 production from this precursor. Comparison of the RT-qPCR generated expression trends for *PRE-MIR399C*, *PRE-MIR399D* and *PRE-MIR399E* in P^+^ and P^−^
*drb4* shoots, to those of P^+^ Col-0, P^−^ Col-0, P^+^
*drb1* and P^−^
*drb1* shoots, revealed a secondary role for DRB4 in regulating miR399 production from these three precursor transcripts (Figure 2C–E). DRB4 also appears to play a role in regulating miR399 production from the *PRE-MIR399F* transcript, with *PRE-MIR399F* abundance reduced by 40% in P^+^
*drb4* shoots (Figure 2F). RT-qPCR also revealed that the expression of this precursor transcript was elevated to a relative expression level of 60.8 in PO_4_-stressed *drb4* shoots; a lower degree of relative expression than observed in either P^−^ Col-0 (92.9 relative expression) or P^−^
*drb1* (178.9 relative expression) shoots (Figure 2F). This finding suggests that in the absence of DRB4 activity, miR399 is more efficiently processed from the *PRE-MIR399F* precursor transcript.

RT-qPCR was next applied to quantify miR399 abundance in the shoot material of non-stressed or PO_4_-stressed Col-0, *drb1*, *drb2* and *drb4* plants. This analysis revealed that in spite of the considerable variation in precursor transcript abundance in the shoot tissues of P^+^ Col-0, P^+^
*drb1*, P^+^
*drb2* and P^+^
*drb4* plants, miR399 levels remained largely unchanged (Figure 2G). This was an especially surprising finding for control *drb1* plants, with the *PRE-MIR399A*, *PRE-MIR399C*, *PRE-MIR399D*, *PRE-MIR399E* and *PRE-MIR399F* transcripts demonstrated to over-accumulate by 4.0-, 10.1-, 12.8-, 5.5- and 14.6-fold in P^+^
*drb1* shoots, compared to their respective levels in P^+^ Col-0 shoots. However, miR399 abundance was only reduced by 10% in P^+^
*drb1* shoots. Similarly, although the expression level of the five miR399 precursors varied considerably in P^+^
*drb2* and P^+^
*drb4* shoots, miR399 abundance was only elevated by 10% and 20%, respectively (Figure 2G). Enhanced miR399 accumulation in P^+^
*drb2* and P^+^
*drb4* shoots did however further identify that both of these DRB proteins are required to correctly regulate miR399 abundance in *Arabidopsis* shoots. The degree of alteration to miR399 abundance was demonstrated to be higher in the shoot tissues of the four assessed plant lines when these lines were cultivated on PO_4_ deplete media. Specifically, RT-qPCR revealed 2.9-, 2.6-, 2.5- and 2.0-fold enhancement to miR399 abundance in PO_4_-stressed Col-0, *drb1*, *drb2* and *drb4* shoots, respectively (Figure 2G).

The mild alteration to miR399 abundance quantified by RT-qPCR in non-stressed and PO_4_-stressed shoots (Figure 2G) led us to next assess the expression of *IPS1*, the eTM of miR399 [32,33,34]. Due to *IPS1* being a PO_4_ stress-induced gene, it was unsurprising to only observe mild (P^+^
*drb2* and P^+^
*drb4* shoots) to moderate differences (P^+^
*drb1* shoots) in *IPS1* transcript abundance in the shoot tissue of non-stressed Col-0, *drb1*, *drb2* and *drb4* plants (Figure 2H). Further, and as expected, RT-qPCR showed that PO_4_ stress induced the expression of *IPS1*, with *IPS1* transcript abundance elevated by 75.7-, 7.1-, 20.8- and 16.4-fold in the shoot tissues of PO_4_ stressed Col-0, *drb1*, *drb2* and *drb4* plants, respectively (compared to the non-stressed counterpart of each plant line).

Next, the expression of the target gene of miR399, *PHO2*, was determined by RT-qPCR to largely remain at wild-type levels (P^+^ Col-0 shoots) in the shoot tissues of P^+^
*drb1*, P^+^
*drb2* and P^+^
*drb4* plants (Figure 2I). This was an unsurprising result considering that RT-qPCR also revealed only mild changes to miR399 abundance across the three *drb* mutant lines assessed when each plant line was cultivated on standard *Arabidopsis* culture media (Figure 2G). RT-qPCR also revealed that elevated miR399 abundance in P^−^ Col-0, P^−^
*drb2* and P^−^
*drb4* plants, promoted miR399-directed expression repression of *PHO2*, with the abundance of the *PHO2* transcript reduced by 50%, 40% and 60% in the shoot tissues of these three plant lines, respectively (Figure 2I). In P^−^
*drb1* shoots however, the level of the *PHO2* transcript was increased by 50% (Figure 2I). Elevated *PHO2* expression in P^−^
*drb1* shoots, a tissue where miR399 abundance was also demonstrated to be elevated, indicated that in the absence of DRB1 activity, miR399-directed mRNA cleavage-mediated regulation of *PHO2* expression is lost.

### 2.3. The Phenotypic and Physiological Response to PO_4_ Stress of the Root System of Arabidopsis Plant Lines Defective in DRB Protein Activity

The unique phenotypic (Figure 1) and molecular (Figure 2) response displayed by *drb1*, *drb2* and *drb4* shoots to PO_4_ starvation led us to next repeat these assessments on the root system of each mutant background. As reported for the aerial tissue phenotypes expressed by the *drb1*, *drb2* and *drb4* mutants (Figure 1), Figure 3A again clearly displays the severe developmental phenotype expressed by the *drb1* mutant as well as the comparatively mild phenotypes that result from the loss of either DRB2 or DRB4 activity in *drb2* and *drb4* plants, respectively. The severity of the developmental phenotypes expressed by the three *drb* mutants assessed in this study is further evidenced when the fresh weight of the root system of 8-day old seedlings cultivated on standard growth media was determined. Specifically, the fresh weight of the root system of 8-day old *drb2* and *drb4* seedlings, 7.95 ± 0.20 mg and 8.00 ± 0.15 mg respectively, was equivalent to the fresh weight of the root system of Col-0 plants, 8.25 ± 0.45 mg (Figure 3B). However, the fresh weight of the root system of 8-day old *drb1* plants, 4.25 ± 0.15 mg, was approximately 50% less than that of an 8-day old Col-0 seedling (Figure 3B).

Figure 3C shows that at the completion of the 7-day PO_4_ starvation period, the fresh weight of 15-day old P^−^ Col-0 roots (29.0 ± 3.0 mg) was only reduced by 2.0 mg compared to P^+^ Col-0 roots (31.0 ± 3.5 mg), a mild 6.5% reduction. The fresh weight of the root system of PO_4_ stressed *drb1*, *drb2* and *drb4* plants all showed a much greater reduction when compared to their non-stressed counterparts (Figure 3C). That is, the fresh weight of the root system of 15-day old P^−^
*drb1* (7.5 ± 0.15 mg), P^−^
*drb2* (23.0 ± 2.5 mg) and P^−^
*drb4* plants (17.5 ± 0.75 mg) was reduced by 25.0%, 25.8% and 18.6%, respectively (Figure 3C).

Inhibition of primary root length is one of the main phenotypic responses of *Arabidopsis* to PO_4_ stress [2,44], and accordingly, Figure 3A,D clearly show that the primary root length of 15-day old P^−^ Col-0 plants (23.4 ± 2.8 mm) was significantly reduced by 51.2% compared to non-stressed P^+^ Col-0 plants (48.1 ± 3.1 mm) (Figure 3D). Although primary root length is already severely inhibited due to detrimental consequences of the loss of DRB1 activity on *Arabidopsis* development, the 7-day stress treatment caused a 46.7% reduction to the primary root length of P^−^
*drb1* plants (10.4 ± 3.1 mm) compared to P^+^
*drb1* plants (19.5 ± 5.9 mm) (Figure 3D). Interestingly, PO_4_ stress impacted primary root development to a much lower degree in both the *drb2* and *drb4* mutant backgrounds. Namely, primary root length was reduced by 20.3% and 10.3% in P^−^
*drb2* (40.5 ± 4.0 mm) and P^−^
*drb4* (41.8 ± 6.2 mm) plants respectively, compared to the primary root length of P^+^
*drb2* (50.8 ± 5.0 mm) and P^+^
*drb4* (46.6 ± 2.9 mm) plants (Figure 3D).

In parallel with inhibition to primary root length, promotion of lateral root development is a commonly reported phenotypic response of *Arabidopsis* plants exposed to PO_4_ stress [2,44]. It was therefore unsurprising to document a 44% increase in the number of lateral roots that formed on 15-day old P^−^ Col-0 plants (4.9 ± 0.4) compared to P^+^ Col-0 plants (3.4 ± 0.3) (Figure 3E). Interestingly, this phenotypic response to PO_4_ stress appeared completely defective in the *drb1* mutant background with both P^+^
*drb1* (4.0 ± 0.2) and P^−^
*drb1* (3.9 ± 0.2) plants forming approximately the same number of lateral roots. Unlike the *drb1* mutant, lateral root development was promoted by ~61% in the *drb2* mutant background with P^−^
*drb2* plants forming 8.2 ± 0.7 lateral roots compared to P^+^
*drb2* plants which formed 5.1 ± 0.8 lateral roots. Lateral root formation was also induced by PO_4_ stress in the *drb4* mutant with the number of lateral roots increased by 44% in P^−^
*drb4* plants (2.6 ± 0.1) compared to their number in P^+^
*drb4* plants (1.8 ± 0.2).

### 2.4. Molecular Profiling of the miR399/PHO2 Expression Module in the Root System of Arabidopsis Plant Lines Defective in DRB Protein Activity

Due to its demonstrated role in inducing *MIR399* gene expression in PO_4_ depleted conditions [27], RT-qPCR was initially used to profile *PHR1* expression in PO_4_-stressed Col-0, *drb1*, *drb2* and *drb4* roots (Figure 4A). This analysis revealed that compared to the root system of each plant line’s non-stressed counterpart, *PHR1* expression remained remarkably constant in P^−^ Col-0, P^−^
*drb1*, P^−^
*drb2* and P^−^
*drb4* roots (Figure 4A). Although RT-qPCR revealed that *PHR1* expression remained constant in the roots of control and PO_4_-stressed plants, RT-qPCR was next applied to profile the expression of the six *MIR399* precursor transcripts in the roots of P^+^ and P^−^ plants. Of the six miR399 precursors, RT-qPCR only allowed for expression quantification of three miR399 precursors, namely *PRE-MIR399A*, *PRE-MIR399C* and *PRE-MIR399D* in *Arabidopsis* roots (Figure 4B–D). In P^−^ Col-0 roots, RT-qPCR clearly revealed that PO_4_ stress induced the expression of the miR399 precursors, *PRE-MIR399A*, *PRE-MIR399C* and *PRE-MIR399D*, by 4.0-, 40.6- and 1546-fold, respectively (Figure 4B–D). When compared to P^+^ Col-0 roots, the moderate 2.3- and 3.6-fold elevation in the abundance of *PRE-MIR399A* and *PRE-MIR399C* in P^+^
*drb1* roots, identified DRB1 as the primary DRB required for miR399 production regulation from these two precursor transcripts in the roots of wild-type *Arabidopsis* plants (Figure 4B,C). The primary role of DRB1 in *PRE-MIR399A* and *PRE-MIR399C* processing in non-stressed Col-0 roots is further evidenced by the wild-type equivalent accumulation of these two precursors in P^+^
*drb2* and P^+^
*drb4* roots, and by the highest degree of *PRE-MIR399A* and *PRE-MIR399C* precursor transcript over-accumulation in P^−^
*drb1* roots (Figure 4B,C). Considering this result, it was therefore of considerable interest to observe the greatest degree of *PRE-MIR399D* over-accumulation, an 8.2-fold increase, in P^+^
*drb4* roots and not in P^+^
*drb1* roots (4.3-fold increase) (Figure 4D). This finding suggests that in non-stressed wild-type *Arabidopsis* roots, DRB4 is the primary DRB responsible for regulating miR399 production from this precursor transcript. In addition, and under PO_4_ stress conditions, the *PRE-MIR399D* transcript increased in its abundance to relative expression values of 829, 849 and 1271 in *drb1*, *drb2* and *drb4* roots, respectively (Figure 4D). Although these determined increases in precursor transcript abundance are all highly significant, they are not as significant as the 1546 relative expression value obtained for the *PRE-MIR399D* transcript in P^−^ Col-0 roots. A lower degree of precursor transcript over-accumulation in each assessed *drb* mutant background, compared to the expression induction observed in wild-type roots, indicated that all three DRB proteins potentially play a role in fine-tuning the regulation of miR399 production from the *PRE-MIR399D* precursor in PO_4_-stressed *Arabidopsis* roots (Figure 4D). 

Post-establishment of highly variable expression profiles for *PRE-MIR399A*, *PRE-MIR399C* and *PRE-MIR399D* in non-stressed *drb1*, *drb2* and *drb4* roots (Figure 4B–D), miR399 abundance reductions of 30%, 50% and 30% in P^+^
*drb1*, P^+^
*drb2* and P^+^
*drb4* roots, respectively, was expected (Figure 4E). Quantification of miR399 abundance, 2.5-, 1.8-, 2.6- and 2.0-fold elevations, respectively, in the root tissues of PO_4_-stressed Col-0, *drb1*, *drb2* and *drb4* plants, revealed that the considerable induction to *PRE-MIR399A*, *PRE-MIR399C* and *PRE-MIR399D* expression (Figure 4B–D), did not however, result in an overly altered miR399 accumulation profile (Figure 4E).

Failure to establish a strong correlation between precursor transcript expression and miR399 abundance in either control or PO_4_-stressed Col-0, *drb1*, *drb2* and *drb4* roots, led us to next assess *IPS1* expression in this tissue (Figure 4F). *IPS1* transcript abundance remained relatively unchanged in the root tissues of non-stressed Col-0 and *drb2* plants (Figure 4F). Interestingly, *IPS1* expression was reduced by 60% in P^+^
*drb1* and P^+^
*drb4* roots (Figure 4F). Significant induction of *IPS1* expression was observed in PO_4_-stressed *drb1*, *drb2* and *drb4* roots, 331-, 696- and 618-fold elevations, respectively. Interestingly, RT-qPCR demonstrated that *IPS1* expression was promoted to its greatest degree, 1076-fold, in PO_4_-stressed Col-0 roots (Figure 4F). 

The expression of the miR399 target gene, *PHO2*, was next quantified by RT-qPCR in non-stressed and PO_4_-stressed Col-0, *drb1*, *drb2* and *drb4* roots (Figure 4G). In P^+^
*drb1* and P^+^
*drb2* roots, RT-qPCR revealed *PHO2* expression to be elevated and reduced by 20%, respectively, and in P^+^
*drb4* roots, *PHO2* expression was reduced by 30%. Elevated *PHO2* expression in P^+^
*drb1* roots was expected considering the slight reduction to miR399 abundance observed in this tissue (Figure 4E). However, the reduced *PHO2* transcript levels in P^+^
*drb2* and P^+^
*drb4* roots was a surprise finding considering that miR399 abundance was also reduced in these two mutant lines by 50% and 30%, respectively (Figure 4E). *PHO2* expression was demonstrated by RT-qPCR to be elevated by 1.9-, 1.6-, 4.5- and 5.1-fold in PO_4_-stressed Col-0, *drb1*, *drb2* and *drb4* roots, respectively (Figure 4G). This finding also formed an unexpected result considering that PO_4_ starvation induced the accumulation of the miR399 sRNA in all four assessed plant lines (Figure 4E). 

### 2.5. Correct Inorganic Phosphate Partitioning Between the Shoot and Root Tissue of Arabidopsis Requires DRB1 and DRB2

The molecular profiling of alterations to the miR399/*PHO2* expression module in the shoot and root tissue of *Arabidopsis* Col-0, *drb1*, *drb2* and *drb4* plants under PO_4_ stress, in combination with each plant line displaying a unique phenotypic response to this stress, led us to next assess Pi partitioning in the aerial tissue and root system of P^+^ and P^−^ Col-0, *drb1*, *drb2* and *drb4* plants. In the shoot tissues of 15-day old plants cultivated in PO_4_ replete conditions, Pi content was only altered in the *drb2* mutant background, with the Pi content of P^+^
*drb2* shoots (13.8 μmol/gFW) reduced by 27.4% compared to the Pi content of P^+^ Col-0 shoots (19.0 μmol/gFW) (Figure 5A). When cultivated in PO_4_-stress conditions however, only the Pi content of P^−^
*drb1* shoots (1.15 μmol/gFW) differed to that of P^−^ Col-0 shoots (1.75 μmol/gFW); a 34.3% reduction (Figure 5A). In non-stressed roots, the Pi content of P^+^
*drb1* (11.4 μmol/gFW) and P^+^
*drb2* (9.8 ± 0.8 μmol/gFW) roots was determined to be elevated by 58.3% and 37.5% respectively, compared to P^+^ Col-0 roots (7.2 μmol/gFW) (Figure 5B). As demonstrated for non-stressed *drb1* and *drb2* roots, the Pi content of P^−^
*drb1* (1.84 μmol/gFW) and P^−^
*drb2* (0.65 μmol/gFW) roots also differed to that of PO_4_-stressed Col-0 roots (1.25 μmol/gFW), elevated and reduced by 47.2% and 48%, respectively (Figure 5B).

The reduced Pi content of P^+^
*drb2* shoots (Figure 5A), together with the elevated Pi contents of P^+^
*drb1* and P^+^
*drb2* roots (Figure 5B), suggested that Pi partitioning was potentially defective in these two mutant backgrounds. We therefore next determined the Pi content ratio of the shoot and root of non-stressed and PO_4_-stressed Col-0, *drb1*, *drb2* and *drb4* plants. Figure 5C clearly shows that Pi partitioning between the shoot and root tissue of P^+^
*drb1* and P^+^
*drb2* plants is defective, even when these two mutant lines are cultivated on standard *Arabidopsis* growth media. Under PO_4_ stress conditions, defective Pi partitioning is even more readily evident in the *drb1* mutant background which showed a 0.38:0.62 shoot to root Pi content ratio, compared to the shoot to root Pi content ratio of 0.58:0.42 for P^−^ Col-0 plants. Although not as striking as determined for P^+^
*drb2* plants, the altered shoot to root Pi content ratio (0.65:0.35) of PO_4_-stressed *drb2* plants again indicated that Pi partitioning is defective in this mutant background (Figure 5D).

Altered shoot to root Pi content ratios in *drb1* and *drb2* plants strongly suggested that Pi partitioning is defective in these two mutant backgrounds. Considering that PO_4_ transporters, PHT1;4, PHT1;8 and PHT1;9, are known targets of PHO2-mediated ubiquitination [7,14], together with our demonstration in Figure 2 and Figure 4 that the miR399/*PHO2* expression module is altered to differing degrees in the shoot and root tissues of each of the three assessed *drb* mutants, RT-qPCR was next applied to profile *PHT1;4*, *PHT1:8* and *PHT1:9* expression in non-stressed and PO_4_-stressed Col-0, *drb1*, *drb2* and *drb4* plants. RT-qPCR revealed that PO_4_ starvation promoted *PHT1;4*, *PHT1:8* and *PHT1:9* expression by 9.1-, 39.6- and 4.3-fold in Col-0 shoots (Figure 5E,G,I), and by 1.2-, 2.6- and 1.4-fold in Col-0 roots, respectively (Figure 5F,H,J). In non-stressed *drb1* shoots, the abundance of the *PHT1;4*, *PHT1:8* and *PHT1:9* transcripts were only mildly altered compared to their respective expression levels in P^+^ Col-0 shoots, returning 1.4-, 1.6- and 2.1-fold changes in expression. A similar mild degree of expression alteration was observed for P^+^
*drb1* roots. Specifically, compared to P^+^ Col-0 roots, the *PHT1;4*, *PHT1:8* and *PHT1:9* transcripts returned fold changes in abundance of 0.6, 1.0 and 0.7, respectively. The expression of these three PO_4_ transporters was significantly induced by the 7-day stress period, returning abundance fold changes of 24.5 (*PHT1;4*), 359.2 (*PHT1:8*) and 242.5 (*PHT1:9*), respectively (Figure 5E,G,I), in P^−^
*drb1* shoots. In spite of the significant induction of *PHT1* gene expression in P^−^
*drb1* shoots, *PHT1;4*, *PHT1:8* and *PHT1:9* levels were reduced (0.7-fold), elevated (2.0-fold) and unchanged (1.0-fold), respectively (Figure 5F,H,J) in the root system of PO_4_-stressed *drb1* roots. As demonstrated for P^+^
*drb1* shoots, RT-qPCR again revealed that *PHT1;4*, *PHT1:8* and *PHT1:9* expression was mildly altered in P^+^
*drb2* shoots by 0.8-, 1.0- and 3.4-fold, respectively. In non-stressed *drb2* roots however, the expression of all three PO_4_ transporters was reduced by 40%, 50% and 60%, respectively, compared to their expression levels in non-stressed Col-0 roots. Furthermore, Figure 5E–J clearly show that the 7-day PO_4_ starvation period induced the expression of these three PO_4_ transporter encoding genes in both the P^−^
*drb2* shoot and root samples, compared to their expression levels in non-stressed *drb2* shoot and roots. Considering that Pi content of non-stressed and PO_4_-stressed *drb4* shoots and roots was determined to be the same as that of the corresponding tissues in P^+^ and P^−^ Col-0 plants, it was unexpected to observe such varied differences in PO_4_ transporter expression across both assessed tissues/growth conditions. For example, in P^+^
*drb4* roots, *PHT1;4*, *PHT1;8* and *PHT1;9* levels were each reduced by 60%, compared to P^+^ Col-0 roots (Figure 5F,H,J), yet the Pi content of non-stressed Col-0 and *drb4* roots was identical (Figure 5B). 

## 3. Discussion

A lack of available P in the soil is a key limitation for plant growth globally [3,45] and as a consequence of P limitation, land plants have evolved highly complex regulatory mechanisms to control both the uptake of external P from the soil, primarily in the form of PO_4_ (Pi), as well as the remobilization of internal stores of P during periods of low external PO_4_ availability [46]. These elaborate P responsive mechanisms allow a plant to attempt to (1) maintain growth and development and (2) regulate cellular P content, regardless of external P concentration [1,2,7]. More contemporary research has focused on the regulatory role played by a suite of PO_4_ responsive miRNA sRNAs that either initiate or maintain PO_4_ signaling pathways across the plant kingdom [4,20]. Central to this PO_4_ responsive miRNA cohort, is miR399, with the miR399 sRNA required to regulate the abundance of the *PHO2* transcript, to in turn regulate the level of the PHO2 protein, an E2 ubiquitin conjugase that mediates the ubiquitin-directed turnover of a group of PO_4_ transporter proteins [7,14,47]. The DRB family members, DRB1, DRB2 and DRB4, have each been ascribed a specific functional role in the *Arabidopsis* miRNA pathway [35,36,37,38,39,40,48,49]. Therefore, we sought to document the involvement of these three DRBs in the production of the PO_4_ responsive miRNA, miR399, and to determine the mode of action directed by the miR399 sRNA during PO_4_ starvation to regulate *PHO2* abundance in the *drb1*, *drb2* and *drb4* mutant backgrounds. Specifically, we attempted to determine what effect an altered miR399/*PHO2* expression module profile would have on the response of *drb1*, *drb2* or *drb4* mutant plants to the imposed stress in order to establish the contribution of either DRB1, DRB2 and/or DRB4 to the maintenance of P homeostasis in *Arabidopsis*.

### 3.1. DRB1 is Required to Maintain Phosphorous Homeostasis in Arabidopsis

Here, it was discovered that the maintenance of P homeostasis is impaired in the *drb1* loss-of-function mutant. The most compelling evidence for this was the documented alteration of the shoot to root Pi content ratio in both non-stressed (Figure 5C) and PO_4_-stressed *drb1* plants (Figure 5D), relative to wild-type *Arabidopsis* (P^+^ or P^−^ Col-0 plants). Specifically, the shoot Pi content was reduced to a much greater degree in PO_4_-stressed *drb1* plants than the observed reduction to Pi content in P^−^ Col-0 shoots. Furthermore, Pi was demonstrated to over-accumulate in the roots of both P^+^ and P^−^
*drb1* plants (Figure 5A,B), compared to the Pi content of the corresponding tissue, and growth regime, of Col-0 plants. The maintenance of appropriate P content in plant tissues is essential for the production of macromolecules, energy trafficking and for numerous signaling pathways [1,2,46]. Therefore, alterations to the P content of the shoot and root tissues of *drb1* plants indicated that in the absence of functional DRB1, P partitioning is impaired. Assessment of the expression of PO_4_ transporters, *PHT1;4*, *PHT1;8* and *PHT1;9*, revealed that the abundance of each transporter was highly elevated by 24.5- 359.2- and 242.5-fold respectively, in the shoot tissue of P^−^
*drb1* plants. Phosphate transporter expression was also demonstrated to be altered in both P^+^ (*PHT1;4* reduced by 1.7-fold and *PHT1;9* reduced by 1.5-fold) and P^−^ (*PHT1;4* reduced by 1.5-fold and *PHT1;8* elevated by 2.0-fold) *drb1* roots, expression alterations that when taken together indicated that incorrect Pi partitioning in *drb1* plants potentially results from defective PO_4_ transport from the root system to the aerial tissue in this mutant background.

Defective root to shoot PO_4_ transport in the *drb1* mutant was further evidenced by the unique phenotypic response displayed by the *drb1* shoot to PO_4_ stress. Specifically, the fresh weight of the shoot of 15-day old P^−^
*drb1* plants was only reduced by 21.6% compared to its non-stressed counterpart (Figure 1C). The rosette area of P^−^
*drb1* plants was also demonstrated to only be reduced by 29.3% post the 7-day PO_4_ stress treatment (Figure 1D). Both responses were comparatively mild compared to the 36.6% and 60.1% reductions to fresh weight and rosette area respectively, documented for Col-0 shoots post the application of PO_4_ stress. In addition, anthocyanin failed to change in abundance in the shoot tissues of P^−^
*drb1* plants compared to the shoots of non-stressed P^+^
*drb1* plants (Figure 1E). Anthocyanin production is a general response to a range of abiotic stresses, including PO_4_ starvation [19,50]. The impaired ability of *drb1* shoots to produce anthocyanin in response to PO_4_ stress may implicate DRB1, and the functional partnership DRB1 forms with DCL1, in the induction of PO_4_ responsive gene expression pathways. Considering these mild responses displayed by *drb1* shoots, it was therefore surprising to observe that chlorophyll *a* and *b* overaccumulation was promoted to the greatest extent in the aerial tissues of *drb1* plants starved of PO_4_. Altered chlorophyll content in P^+^
*drb1* shoots indicated that (1) *drb1* shoots are indeed negatively impacted by the imposed PO_4_ stress, and (2) that DRB1 may potentially mediate a PO_4_-directed role in regulating photosynthesis in *Arabidopsis* chloroplasts.

Considering the well-established role of the DRB1/DCL1 functional partnership in the production of the majority of miRNAs that accumulate in *Arabidopsis* tissues, it was unsurprising to observe that the miR399 precursors, *PRE-MIR399A*, *PRE-MIR399C*, *PRE-MIR399D*, *PRE-MIR399E* and *PRE-MIR399F*, over-accumulated to the greatest extent in P^+^
*drb1* shoots (Figure 2A–E). In addition, precursors *PRE-MIR399A*, *PRE-MIR399C*, *PRE-MIR399D* and *PRE-MIR399F* were further demonstrated to be most highly abundant in the shoot tissues of PO_4_-stressed *drb1* plants. The enhanced abundance of miRNA precursor transcripts in the *drb1* mutant background is most likely the result of inefficient precursor transcript processing by DCL1 in the absence of DRB1 functional assistance, with DRB1 accurately positioning DCL1 on each miRNA precursor to direct accurate processing [48,49]. In spite of the readily observable evidence of inefficient miR399 precursor transcript processing in P^+^
*drb1* shoots, miR399 levels were only reduced by 10% (Figure 5G). Similarly, although miR399 precursor transcript abundance was elevated to a much greater degree in P^−^
*drb1* shoots due to a combination of (1) *MIR399* gene expression induction in response to PO_4_ starvation, and (2) inefficient precursor transcript processing in the absence of DRB1 activity, miR399 abundance was again demonstrated to be only mildly elevated by 2.3-fold in the shoots of PO_4_-stressed *drb1* plants (Figure 5G). Further, the abundance of the miR399 target transcript, *PHO2*, was only mildly elevated by 1.2-fold in response to the 10% reduction in miR399 levels in P^+^
*drb1* shoots (Figure 2I). Surprisingly, *PHO2* transcript abundance was elevated by 1.5-fold in response to the 2.3-fold elevation in miR399 accumulation in P^−^
*drb1* shoots, and not reduced as expected. However, in P^+^ Col-0 shoots, and as expected, the 2.9-fold enhancement to miR399 abundance led to a 50% reduction in *PHO2* expression (Figure 5G,I). Therefore, elevated *PHO2* abundance in response to enhanced miR399 levels in P^−^
*drb1* shoots, readily demonstrates that miR399-directed *PHO2* transcript cleavage, to regulate *PHO2* expression, is defective in the absence of DRB1 activity.

Altered PO_4_ transporter expression in *drb1* roots indicated that the response of the root system of the *drb1* mutant to PO_4_ stress would differ to that of the root system of wild-type *Arabidopsis*. Accordingly, the fresh weight of PO_4_-stressed *drb1* roots was reduced by 25.0% compared to the mild 6.5% reduction to the fresh weight of P^−^ Col-0 roots, a 3.8-fold enhancement to the severity of this phenotypic response (Figure 3C). It was therefore curious to observe a similar degree of reduction to primary root length in P^−^
*drb1* (46.7%) and P^−^ Col-0 (51.2%) plants (Figure 3D). A greater degree of reduction to the fresh weight of P^−^
*drb1* roots, compared to P^−^ Col-0 roots, could be partially explained by the observation that the induction of lateral root formation by PO_4_ stress was completely defective in P^−^
*drb1* roots, compared to a 44.0% increase in lateral root number in P^−^ Col-0 roots (Figure 3D). Considering that the measurement of fresh weight is largely assessing the moisture content of a plant, the observed reduction to fresh weight of P^−^
*drb1* roots could potentially be indicating that under PO_4_ stress conditions, DRB1 is somehow involved in regulating the moisture content of the root system of *Arabidopsis*. However, this was not assessed in this study with the mechanism driving the enhancement of fresh weight reductions requiring further investigation in the future.

Similar to its establishment as the primary DRB protein required to regulate miR399 production from the *PRE-MIR399A*, *PRE-MIR399C*, *PRE-MIR399D*, *PRE-MIR399E* and *PRE-MIR399F* precursors in the aerial tissues of non-stressed *Arabidopsis* plants, DRB1 was again demonstrated to be the primary DRB protein required to regulate miR399 production from the *PRE-MIR399A* and *PRE-MIR399C* precursor transcripts in the *Arabidopsis* root system with both precursors demonstrated to accumulate to the greatest degree in P^+^ and P^−^
*drb1* roots (Figure 4B,C). Reduced *PRE-MIR399A* and *PRE-MIR399C* processing efficiency in the absence of DRB1 activity, reduced miR399 abundance by 30% in P^+^
*drb1* roots (Figure 4E), and in turn, this moderate reduction to miR399 levels led to a mild elevation (1.2-fold) in the expression of the miR399 target gene, *PHO2* (Figure 4G). As documented in P^−^
*drb1* shoots, the 1.8-fold elevation to miR399 levels in P^−^
*drb1* roots, resulted in a moderate elevation to *PHO2* transcript abundance (1.6-fold), and not a reduction in target gene expression as would be expected for a miRNA that regulates the expression of its targeted genes solely via a mRNA cleavage mode of RNA silencing. However, considering that a similar miRNA/target gene expression profile of elevated miR399 abundance (2.5-fold), together with enhanced *PHO2* expression (1.9-fold) was also observed in PO_4_-stressed Col-0 roots, this curious finding indicates that miR399-directed *PHO2* transcript cleavage may not be the predominant mechanism of target gene expression regulation directed by the miR399 sRNA in the *Arabidopsis* root system. Alternatively, elevated *PHO2* expression in P^+^ Col-0 and P^+^
*drb1* roots when miR399 abundance is also elevated may result from the enhanced expression of the eTM of miR399 activity, *IPS1*. In P^−^ Col-0 shoots for example, where elevated miR399 abundance was demonstrated to direct enhanced expression repression of the *PHO2* transcript (Figure 2G,I), *IPS1* abundance was elevated by 75.7-fold, compared to its abundance in P^+^ Col-0 shoots (Figure 2H). In PO_4_-stressed roots, however, *IPS1* expression was elevated to a much greater degree, by 1076-fold (Figure 4F). This 14.2-fold greater promotion to *IPS1* expression in P^−^ Col-0 roots, than that observed in P^−^ Col-0 shoots, would be expected to sequester a higher amount of miR399, which in turn, could have led to the observed elevation in *PHO2* expression in P^−^ Col-0 roots in the presence of 2.5-fold greater abundance of the *PHO2* targeting miRNA, miR399.

### 3.2. DRB2 is Required to Maintain Phosphate Homeostasis in Arabidopsis

As documented for the *drb1* mutant, P homeostasis was determined to be defective in the *drb2* mutant. Specific to *drb2* plants however, was the 27.8% reduction to the Pi content of non-stressed *drb2* shoots (Figure 5A). Of the four *Arabidopsis* plant lines assessed in this study, *drb2* was the only line determined to have a reduced aerial tissue Pi content when cultivated under standard growth conditions. Furthermore, in P^+^
*drb2* shoots, *PHT1;4* (Figure 5E) and *PHT1;8* (Figure 5G) expression was determined to be reduced and elevated by 1.2- and 3.4-fold respectively, compared to the expression of these two PO_4_ transporters in P^+^ Col-0 shoots. In addition, Pi was determined to over-accumulate by 36.1% in P^+^
*drb2* roots. In P^+^
*drb2* roots, *PHT1;4*, *PHT1;8* and *PHT1;9* expression was reduced by 1.7-, 2.0- and 2.4-fold respectively, compared to their expression levels in P^+^ Col-0 roots. Together, (1) reduced Pi content of P^+^
*drb2* shoots, (2) elevated Pi content in P^+^
*drb2* roots, and (3) reduced PO_4_ transporter gene expression in P^+^
*drb2* roots, indicated that PO_4_ root to shoot transport is defective in non-stressed *drb2* plants. Based on this finding, it was curious to observe a similar Pi content in P^−^
*drb2* shoots and P^−^ Col-0 shoots (Figure 5A), especially considering the document enhancement to *PHT1;4* and *PHT1;9* expression in P^−^
*drb2* shoots, with the expression of these two PO_4_ transporters elevated by 2.8- and 7.0-fold respectively, compared to the degree of expression induction observed in P^−^ Col-0 roots (Figure 5E,I). However, and as demonstrated for P^+^
*drb2* shoots and roots, the Pi content of the root system of PO_4_-stressed *drb2* plants was altered, reduced by 48% compared to the Pi content of P^−^ Col-0 roots. Interestingly, RT-qPCR revealed similar degrees of elevated *PHT1;8* (Figure 5H) and *PHT1;9* (Figure 5J) expression in PO_4_-stressed Col-0 and *drb2* roots with only the *PHT1;4* transcript returning a slight difference in its expression in P^−^ Col-0 roots (elevated by 1.2-fold compared to P^+^ Col-0 roots) and P^−^
*drb2* roots (reduced by 1.1-fold compared to P^+^ Col-0 roots). The PO_4_ transporters, PHT1;1 and PHT1;4, have been demonstrated to be responsible for the import of more than half of the Pi that is taken up from the soil [51]. It therefore seems unlikely that the mild 10% reduction to *PHT1;4* transcript abundance documented in PO_4_-stressed *drb2* roots, is the sole cause of the considerable reduction to the Pi content of the root system in the *drb2* mutant background.

Considering that the Pi content of PO_4_-stressed Col-0 and *drb2* shoots was determined to be similar, it was unsurprising to document a similar degree of reduction to fresh weight of the shoot tissues of P^−^ Col-0 (36.6%) and P^−^
*drb2* (39.1%) plants (Figure 1C). Rosette area was also decreased by a similar degree in P^−^ Col-0 (60.1%) and P^−^
*drb2* (48.0%) plants (Figure 1D). However, compared to PO_4_-stressed Col-0 shoots, anthocyanin accumulated to considerably higher levels in the aerial tissues of *drb2* plants when exposed to PO_4_ stress (Figure 1E). The induction of anthocyanin production is a well-characterized response to PO_4_ starvation [19,50]. Therefore, the considerable enhancement of anthocyanin accumulation in P^−^
*drb2* shoots, compared to the shoot tissues of PO_4_-stressed Col-0 plants, suggests that this P-responsive pathway is hyperactivated in the absence of DRB2 activity, as well as potentially implicating DRB2 in mediating a regulatory role in a range of other P-responsive pathways in *Arabidopsis* aerial tissues that were not assessed in this study.

We have previously demonstrated a role for DRB2 in the production stage of the *Arabidopsis* miRNA pathway with the abundance of specific miRNA cohorts altered in the *drb2* mutant background [37]. More specifically, DRB2 can either be antagonistic or synergistic to DRB1 function in the DRB1/DCL1 partnership for the production of specific miRNAs, resulting in miRNA abundance either being enhanced (antagonistic) or reduced (synergistic) in *drb2* plants [37,38]. Reduced precursor transcript abundance in non-stressed *drb2* shoots, indicated that DRB2 plays a secondary role in regulating miR399 production from the *PRE-MIR399A*, *PRE-MIR399E* and *PRE-MIR399F* precursors, potentially via antagonism of DRB1 function (Figure 2B,E,F). The antagonism of DRB2 on the DRB1/DCL1 partnership becomes more readily apparent via the profiling of miR399 precursor transcript expression in P^−^
*drb2* shoots, with lower degrees of expression induction observed for the *PRE-MIR399C*, *PRE-MIR399D*, *PRE-MIR399E* and *PRE-MIR399F* precursors (Figure 2C–F). Reduced precursor transcript abundance in P^−^
*drb2* shoots, compared to the respective abundance of each precursor in either P^−^ Col-0 or P^−^
*drb1* shoots, indicates that in the absence of DRB2 activity, precursor transcript processing efficiency is enhanced due to more precursor transcript being freely available to enter the canonical DRB1/DCL1 production pathway. 

As demonstrated in P^+^
*drb1* shoots, significantly altered precursor transcript abundance in P^+^
*drb2* shoots, failed to have a strong influence on the accumulation of miR399, with miR399 levels only mildly elevated by 10% in P^+^
*drb2* shoots, compared to P^+^ Col-0 shoots (Figure 5G). However, DRB2 antagonism was still evidenced by this mild increase to miR399 abundance compared to the 10% reduction in miR399 levels observed in P^+^
*drb1* shoots. The antagonism of DRB2 on miR399 production was further evidenced by the enhanced expression repression of *PHO2* in P^−^
*drb2* shoots (Figure 2I). The abundance of miR399 was elevated by 2.7-fold in P^−^
*drb2* shoots, and therefore, a further degree of reduced *PHO2* expression in P^−^
*drb2* shots, compared to P^−^
*drb1* shoots where miR399 levels were elevated by 2.3-fold and *PHO2* expression was enhanced by 1.5-fold, clearly demonstrated enhanced DRB1-mediated, miR399-directed, *PHO2* transcript cleavage in the absence of DRB2 antagonism. Similarly, it is important to note here that *IPS1* transcript abundance was enhanced to a much lower degree in P^−^
*drb2* shoots (27.1-fold) compared to *IPS1* abundance induction in either PO_4_-stressed Col-0 (75.7-fold) or *drb1* (85.4-fold) shoots. This unexpected observation again indicated that in the absence of DRB2 activity, miR399-directed target transcript cleavage was enhanced. Although *IPS1* has been identified as a non-cleavable eTM of miR399 activity, the *IPS1* expression trends presented in Figure 5H suggest that miR399 may well be capable of directing miRISC-catalyzed cleavage of the *IPS1* transcript in addition to solely being sequestered by *IPS1*.

Compared to the mild 6.5% reduction to the fresh weight of P^−^ Col-0 roots, the negative response of the root system of the *drb2* mutant to PO_4_ stress was considerably more pronounced at 25.8% (Figure 3C). Considering that the correct regulation of Pi content is dysfunctional in both control and PO_4_-stressed *drb2* roots, differing responses to PO_4_ stress in *drb2* roots, compared to P^−^ Col-0 roots, was not surprising. Similarly, inhibition of the primary root length of P^−^
*drb2* plants at 20.3% was comparatively mild compared to the severe 51.2% inhibition to the primary root length observed for P^−^ Col-0 plants (Figure 5D). The degree of lateral root induction also differed between PO_4_-stressed Col-0 and *drb2* roots (Figure 5E), specifically; lateral root formation was enhanced by ~44% in P^−^ Col-0 plants, and in PO_4_-stressed *drb2* plants, lateral root formation was further promoted by 17% with P^−^
*drb2* plants developing ~61% more lateral roots than their non-stressed counterparts. When these phenotypic responses of the root system of PO_4_-stressed *drb2* plants are considered together, including a lower degree of primary root length inhibition (2.5-fold less than P^−^ Col-0 plants), and a more pronounced enhancement to lateral root formation (1.4-fold more than P^−^ Col-0 plants), it was highly surprising that the fresh weight of P^−^
*drb2* roots was reduced by a 4.0-fold greater degree than documented for P^−^ Col-0 roots.

Similar levels of expression of *PRE-MIR399A* in both non-stressed and PO_4_-stressed Col-0 and *drb2* roots revealed that DRB2 does not play a role in regulating miR399 processing from this precursor transcript (Figure 4B). Reduced expression of *PRE-MIR399C* in P^+^
*drb2* roots (compared to P^−^ Col-0 and P^−^
*drb1* roots) and a lower level of precursor over-accumulation in P^−^
*drb2* roots (compared to P^−^
*drb1* roots), identified DRB2 as playing a secondary role in regulating miR399 production from this precursor transcript in the *Arabidopsis* root system (Figure 4C) via antagonism of DRB1 function. The expression trend of *PRE-MIR399D* in P^−^
*drb2* roots additionally identified a secondary role for DRB2 in regulating miR399 production from the third miR399 precursor transcript detected in the root system of the four *Arabidopsis* plants lines assessed in this study. However, for the *PRE-MIR399D* precursor, DRB2 appears to be antagonistic to the DRB4/DCL4 partnership, and not to the canonical DRB1/DCL1 partnership demonstrated to be required for the production of the majority of *Arabidopsis* miRNAs. DRB2 has been demonstrated previously to be antagonistic to DRB4 function in the DRB4/DCL4 partnership for the production of a small subset of newly evolved *Arabidopsis* miRNAs processed from precursor transcripts that fold to form highly complementary stem-loop structures [39,40]. Considering that in P^+^
*drb2* roots, *PRE-MIR399A* and *PRE-MIR399D* remained at their approximate wild-type levels, and that the *PRE-MIR399C* precursor was reduced in its abundance by 1.7-fold, a finding that initially indicated that this precursor is more efficiently processed by DRB1/DCL1 in the absence of DRB2 activity, the 2.0-fold reduction to miR399 abundance alternatively indicated that *MIR399C* gene expression may in fact be reduced in PO_4_-stressed *drb2* roots. It was therefore curious to observe *PHO2* expression to be reduced by 1.3-fold in P^+^
*drb2* roots, and not elevated in response to reduced miR399 abundance as expected. However, this observation is potentially demonstrating that in spite of being reduced in abundance, this lower level of miR399 directs more efficient cleavage of the *PHO2* transcript in the absence of DRB2 activity. In P^−^
*drb2* roots, miR399 abundance was determined to be elevated by 2.6-fold compared to its abundance in P^−^ Col-0 roots (Figure 4E). As observed in P^+^
*drb2* roots, *PHO2* expression scaled in accordance with elevated miR399 abundance, with *PHO2* expression increased by 4.5-fold in PO_4_-stressed *drb2* roots. It is interesting to note here that *PHO2* expression scaled with miR399 abundance in six out the eight root tissue samples molecularly assessed by RT-qPCR in this study. We have previously demonstrated that DRB2-dependent miRNAs direct a translational repression mode of miRNA-directed target gene expression repression [52], and scaling of miRNA target transcripts together with their targeting miRNA, has been previously reported for miRNA sRNAs that direct a translational repression mode of target gene expression regulation [52,53,54].

### 3.3. DRB4 is Required For miR399 Production in Arabidopsis Roots

Profiling of PO_4_ transporter expression in the shoots and roots of P^+^ and P^−^
*drb4* plants revealed considerable alteration to *PHT1;4*, *PHT1;8* and *PHT1;9* transcript abundance across both assessed tissues and growth regimes (Figure 5E–J). However, in spite of these documented differences in PO_4_ transporter gene expression in *drb4* shoots and roots, the Pi content of non-stressed and PO_4_-stressed *drb4* tissues remained at levels comparable to P^+^ and P^−^ Col-0 shoots and roots (Figure 5A,B). Considering this finding, it was unsurprising that the developmental progression of Col-0 and *drb4* plants was impeded to the same extent when cultivated in the absence of PO_4_ for a 7-day period. Specifically, the fresh weight of both P^−^ Col-0 and P^−^
*drb4* shoots was reduced by ~36% compared to their non-stressed counterparts of the same age (Figure 1C). In addition, anthocyanin, chlorophyll *a* and chlorophyll *b* were all elevated to the same degree in PO_4_-stressed Col-0 and *drb4* shoots, compared to their respective non-stressed counterparts. It was therefore surprising that the rosette area of P^−^
*drb4* plants was only reduced by 38.7% compared to the more severe 60.1% reduction observed for P^−^ Col-0 plants. Although an unexpected finding, this result clearly indicated that some of the responses of the *drb4* mutant to PO_4_ starvation differ to those of wild-type *Arabidopsis*.

Considering the well-established role of the DRB4/DCL4 partnership in *trans*-acting siRNA (tasiRNA) [55,56] and p4-siRNA [40] production, and for the processing of a small number of newly evolved miRNAs from their highly complementary precursor transcripts [39], it was highly surprising to additionally establish the widespread involvement of DRB4 in regulating the production of the highly conserved miRNA, miR399, in *Arabidopsis* shoots (Figure 2). Specifically, DRB4 was determined to play a secondary role to DRB1 in regulating the efficiency of miR399 production from all five precursors detectable by RT-qPCR in non-stressed *Arabidopsis* shoots. As demonstrated for DRB2, the involvement of DRB4 in miR399 production in *Arabidopsis* shoots is most likely via antagonism of the canonical DRB1/DCL1 partnership. Antagonism of the DRB1/DCL1 partnership by DRB4 was again demonstrated by the accumulation profiles of precursors, *PRE-MIR399C*, *PRE-MIR399D*, *PRE-MIR399E* and *PRE-MIR399F*, in the shoot tissues of PO_4_-stressed *drb4* plants (Figure 2C–F). Although precursor transcript abundance was highly variable in *drb4* shoots, miR399 levels were only mildly elevated by 1.2- and 2.4-fold in P^+^
*drb4* and P^−^
*drb4* shoots, respectively (Figure 2G). Surprisingly, in spite of the 20% elevation to miR399 levels in P^+^
*drb4* shoots, *PHO2* expression was elevated to a similar degree (30% increase), and not reduced as expected (Figure 5I). In P^−^
*drb4* shoots, however, the 2.4-fold elevated abundance of the miR399 sRNA was determined, as expected, to reduce the expression of *PHO2* by 2.5-fold. This result clearly indicated that in the absence of DRB4 activity in *Arabidopsis* shoots, the efficiency of DRB1-mediated, miR399-directed cleavage of the *PHO2* transcript is enhanced.

The fresh weight of P^−^
*drb4* roots was reduced by 18.6% compared to the fresh weight of P^+^
*drb4* roots, a 2.9-fold further enhancement of this phenotypic response to PO_4_ stress, compared to the mild response of P^−^ Col-0 roots (6.5% fresh weight reduction compared to P^+^ Col-0 roots). The response of the primary root of the *drb4* mutant to PO_4_ stress also differed to that of wild-type roots. Namely, the length of P^−^
*drb4* primary root was only reduced by 10.3% compared to the significant 51.2% reduction to the length of the primary root of P^−^ Col-0 plants (Figure 3D). Although lateral root development was induced to the same degree (44%) in the root system of PO_4_-stressed Col-0 and *drb4* plants, the considerable differences observed for the fresh weight of the *drb4* root system, and the lack of inhibition to primary root length in P^−^
*drb4* plants, clearly revealed that the *drb4* mutant background is defective in some of its responses to PO_4_ starvation, compared to the responses of the Col-0 root system to this stress.

At the molecular level, the wild-type-like expression of the *PRE-MIR399A* precursor in the roots of non-stressed and PO_4_-stressed *drb4* plants indicated that DRB4 does not play a role in regulating miR399 production from this precursor in *Arabidopsis* roots. Expression analysis of *PRE-MIR399C* did however identify a secondary role for DRB4 in regulating miR399 production from this precursor, potentially via antagonism of DRB1 function (Figure 4C). Of particular interest stemming from miR399 precursor transcript profiling in non-stressed and PO_4_-stressed *Arabidopsis* roots is the unexpected finding that DRB4 appears to be the primary DRB required to regulate miR399 production from the *PRE-MIR399D* precursor (Figure 4D), with the abundance of the *PRE-MIR399D* precursor over-accumulating to its highest levels in both P^+^ and P^−^
*drb4* roots. Curiously, assessment of the stem-loop folding structures of the six precursors from which the miR399 sRNA is liberated does not readily distinguish the *PRE-MIR399D* structure from the folding structures of the other five miR399 precursor transcripts. Therefore, the establishment of a role for DRB4 in regulating miR399 processing efficiency from its precursor transcripts was a highly unexpected finding, a finding that requires additional experimentation in the future to identify the precursor transcript-based sequence and/or structural features that recruits the involvement of DRB4 to the miR399/*PHO2* expression module. 

The elevated abundance of the *PRE-MIR399C* and *PRE-MIR399D* precursors in P^+^
*drb4* roots indicated reduced precursor transcript processing efficiency in the absence of DRB4. Accordingly, a 30% reduction to miR399 accumulation was observed in P^+^
*drb4* roots (Figure 4E). Surprisingly, this 1.4-fold reduction to miR399 levels in P^+^
*drb4* roots led to a 2.0-fold reduction to *PHO2* expression (Figure 4G). This result suggested that although miR399 levels were reduced in non-stressed *drb4* roots, the reduced amount of the miR399 sRNA was actually directing enhanced *PHO2* expression repression via unimpeded DRB1-mediated, miR399-directed, *PHO2* cleavage. However, enhanced miR399-directed *PHO2* cleavage appeared to be lost in PO_4_-stressed *drb4* roots with both miR399 and *PHO2* levels elevated by 2.0- and 5.1-fold, respectively (Figure 4E,G). Therefore, when taken together, although miR399-directed *PHO2* cleavage appeared to be enhanced in P^+^
*drb4* roots, the scaling of *PHO2* expression together with miR399 abundance in PO_4_-stressed *drb4* roots, potentially suggests that in a cell type with altered physiology, and where DRB4 function is defective, the miR399 sRNA changed from directing an mRNA cleavage mode of RNA silencing, to directing a translational repression mode of RNA silencing.

### 3.4. DRB1, DRB2 and DRB4 Are Required to Regulate the miR399/PHO2 Expression Module in Arabidopsis Shoots and Roots

Here we demonstrate that the phenotypic and molecular response to PO_4_ starvation were unique to each *drb* mutant background assessed due to the hierarchical contribution of DRB1, DRB2 and DRB4 to the regulation of the miR399/*PHO2* expression module. Specifically, the molecular profiling of miR399 precursor transcript expression identified DRB1 as the primary DRB required for efficient miR399 production from each precursor in non-stressed and PO_4_-stressed shoots and roots. Deregulated miR399 precursor transcript processing efficiency in the absence of DRB1 activity was demonstrated to result in defective P homeostasis maintenance, altering the shoot to root ratio of Pi content in the *drb1* mutant background. The maintenance of P homeostasis was also defective in *drb2* plants, with the Pi content shoot to root ratio altered in this mutant background, both under standard growth conditions and in conditions of PO_4_ starvation. An altered Pi content in *drb2* tissues appeared to result from defective PO_4_ transport between the root system and aerial tissues in the absence of DRB2 function. Further, DRB2 was determined to play a secondary role to DRB1 in regulating miR399 production from the profiled *PRE-MIR399* precursor transcripts. The secondary role of DRB2 in regulating miR399 production from the assessed *PRE-MIR399* precursor transcripts was revealed to most likely be via antagonism of DRB1 function. DRB4 was also determined to play a secondary role in regulating the miR399/*PHO2* expression module in *Arabidopsis* shoots and roots, and as demonstrated for the secondary role of DRB2 in providing additional regulatory complexity to this expression module, DRB4 also appeared to be antagonistic to the primary functional role of DRB1 in regulating miR399 precursor transcript processing efficiency. Furthermore, DRB4 also appeared to be the primary DRB required for miR399 production from the *PRE-MIR399D* precursor in non-stressed and PO_4_-stressed *Arabidopsis* roots. When taken together, the hierarchical contribution of DRB1, DRB2 and DRB4 to the regulation of the miR399/*PHO2* expression module documented here, readily demonstrates the crucial importance of maintaining P homeostasis in *Arabidopsis* tissues to ensure the maintenance of a wide range of cellular processes to which P is essential.

## 4. Materials and Methods

### 4.1. Plant Material and Phosphate Stress Treatment

The T-DNA insertion knockout mutant lines used in this study, including the *drb1* (*drb1-1*; SALK_064863), *drb2* (*drb2-1*; GABI_348A09) and *drb4* (*drb4-1*; SALK_000736) mutants, have been described previously [42]. The seeds of these three *drb* mutant lines, and of wild-type *Arabidopsis* (ecotype Columbia-0 (Col-0)) plants, were sterilized using chlorine gas and post-sterilization, seeds were plated out onto standard *Arabidopsis* plant growth media (half-strength Murashige and Skoog (MS) salts), and stratified for 48 h at 4 °C in the dark. Post-stratification, the sealed plates were transferred to a temperature-controlled growth cabinet (A1000 Growth Chamber, Conviron^®^ Australia) and cultivated for an 8-day period under a standard growth regime of 16 h light / 8 h dark, and a day/night temperature of 22 °C / 18 °C. Post this initial 8-day cultivation period, equal numbers of Col-0, *drb1*, *drb2* and *drb4* seedlings were transferred under sterile conditions to either fresh standard *Arabidopsis* plant growth media that contained 1.0 mM of PO_4_ (P^+^ plants; non-stressed controls) or to *Arabidopsis* plant growth media where the PO_4_ had been replaced with an equivalent molar amount (1.0 mM) of potassium chloride (KCl) (P^−^ plants; PO_4_ stress treatment). Post seedling transfer, the P^+^ and P^−^ plates for each plant line were returned to the temperature-controlled growth cabinet for an additional 7-day cultivation period. For the tissue-specific analyses performed here, namely the root tissue assessments, additional Col-0, *drb1*, *drb2* and *drb4*, 8-day old seedlings were treated exactly as outlined above, except for the 7-day treatment period, where P^+^ and P^−^ plates were orientated for vertical growth. Unless stated otherwise, all the phenotypic and molecular analyses reported here were conducted on 15-day old plants.

### 4.2. Phenotypic and Physiological Assessments

The fresh weight of 8-day old Col-0, *drb1*, *drb2* and *drb4* whole plants germinated and cultivated on standard *Arabidopsis* plant growth media was initially determined to establish the effect of loss of DRB1, DRB2 or DRB4 activity on *Arabidopsis* development. The fresh weight of 15-day old Col-0, *drb1*, *drb2* and *drb4* plants was also determined to establish the effect of the 7-day PO_4_ stress treatment on the development of each plant line. The area of the rosette and the length of the primary root of 15-day old Col-0, *drb1*, *drb2* and *drb4* plants was determined via the assessment of photographic images using the ImageJ software. The same photographic images were also used to establish the number of lateral roots formed by P^+^ and P^−^ Col-0, *drb1*, *drb2* and *drb4* plants post the 7-day stress treatment period. 

A standard methanol:HCl (99:1 v/v) extraction method was applied to extract anthocyanin from P^+^ and P^−^ plants, and post extraction, anthocyanin content was determined using a spectrophotometer (Thermo Scientific, Australia) at an absorbance wavelength of 535 nanometers (A_535_). The 99:1 (v/v) methanol:HCl extraction buffer was used as the blanking solution and the A_535_ of each sample was next divided by the fresh weight of the sample to calculate the relative anthocyanin content per gram of fresh weight (A_535_/g FW).

For chlorophyll *a* and *b* content quantification, rosette leaves of 15-day old P^+^ and P^−^ Col-0, *drb1*, *drb2* and *drb4* plants were sampled and incubated in 80% acetone for 24 h in the dark. Post incubation, rosette leaf tissue was clarified via centrifugation at 15,000 × *g* for 7 min at room temperature. The resulting supernatants were immediately transferred to a spectrophotometer and the absorbance of these solutions assessed at wavelengths 646 nm (A_646_) and 663 nm (A_663_) using 80% acetone as the blanking solution. The chlorophyll *a* and *b* content of each sample was then determined using the Lichtenthaler’s equations exactly as outlined in [57], and these initially determined values were subsequently converted to micrograms per gram of fresh weight (μg/g FW).

The shoot and root tissue of 15-day old P^+^ and P^−^ Col-0, *drb1*, *drb2* and *drb4* plants were carefully separated from each other and then ground into a fine powder under liquid nitrogen (LN_2_). One milliliter (1.0 mL) of 10% acetic acid (v/v in H_2_O) was added to the ground powder and the powder thoroughly resuspended via vigorous vortexing. The resulting resuspension was then centrifuged at 15,000 × *g* for 5 min at room temperature, and post centrifugation, 700 μL of the resulting supernatant was mixed with an equivalent volume of Ames Assay Buffer (6 parts 0.5% ammonium molybdite (v/v in H_2_O) to 1 part of 2.5% sulphuric acid (v/v in 10% acetic acid)) and incubated at room temperature for 1 h in the dark. The absorbance of each solution was determined using a spectrophotometer at wavelength 820 nm (A_820_) and the Pi content (μmol/gFW) of each sample subsequently determined via the construction of a Pi standard curve.

### 4.3. Total RNA Extraction for Quantitative Molecular Assessments

For each molecular assessment reported here, total RNA was extracted from four biological replicates (each biological replicate contained tissue sampled from eight individual plants) of 15-day old P^+^ and P^−^ Col-0, *drb1*, *drb2* and *drb4* plants using TRIzol^TM^ Reagent according to the manufacturer’s (Invitrogen^TM^) instructions. The quality of the extracted total RNA was visually assessed via a standard electrophoresis approach on a 1.2% (w/v) ethidium bromide stained agarose gel and the quantity of total RNA extracted determined using a NanoDrop spectrophotometer (NanoDrop^®^ ND-1000, Thermo Scientific, Australia).

For the synthesis of a miR399-specific complementary DNA (cDNA), 200 nanograms (ng) of total RNA was treated with 0.2 units (U) of DNase I (New England Biolabs, Australia) according to the manufacturer’s instructions. The DNase I-treated total RNA was next used as template for cDNA synthesis with 1.0 U of ProtoScript^®^ II Reverse Transcriptase (New England Biolabs, Australia) and the cycling conditions of 1 cycle of 16 °C for 30 min; 60 cycles of 30 °C for 30 s, 42 °C for 30 s, and 50 °C for 2 s, and; 1 cycle of 85 °C for 5 min.

A global, high molecular weight cDNA library for gene expression quantification was constructed via the initial treatment of 5.0 μg of total RNA with 5.0 U of DNase I according to the manufacturer’s protocol (New England Biolabs, Australia). The DNase I-treated total RNA was next purified using an RNeasy Mini Kit (Qiagen, Australia) and 1.0 μg of this preparation used as template for cDNA synthesis along with 1.0 U of ProtoScript^®^ II Reverse Transcriptase (New England Biolabs, Australia) and 2.5 mM of oligo dT_(18)_, according to the manufacturer’s instructions.

All generated, single-stranded cDNAs were next diluted to a working concentration of 50 ng/μL in RNase-free H_2_O prior to their use as a template for the quantification of the abundance of either the miR399 sRNA or of gene transcripts. In addition, all RT-qPCRs used the same cycling conditions of 1 cycle of 95 °C for 10 min, followed by 45 cycles of 95 °C for 10 s and 60 °C for 15 s, and the GoTaq^®^ qPCR Master Mix (Promega, Australia) was used as the fluorescent reagent for all performed RT-qPCR experiments. miR399 abundance and gene transcript expression was quantified using the 2^−^^ΔΔCT^ method with the small nucleolar RNA, *snoR101*, and *UBIQUITIN10* (*UBI10*; *AT4G05320*) used as the respective internal controls to normalize the relative abundance of each assessed transcript. The sequence of each DNA oligonucleotide used in this study either for the synthesis of a miR399-specific cDNA, or to quantify transcript abundance via RT-qPCR is provided in Appendix A. 

## Figures and Tables

**Figure 1 plants-08-00124-f001:**
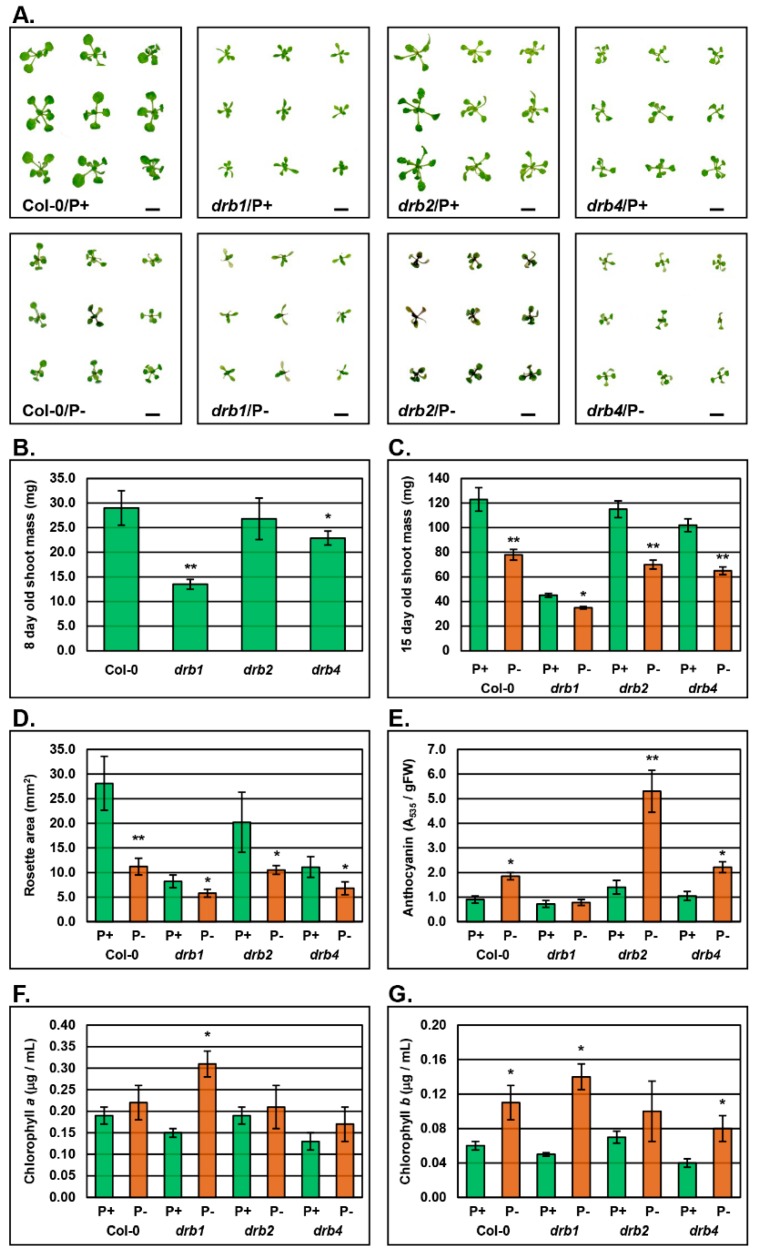
The aerial tissue phenotypes displayed by 15-day old *Arabidopsis* plant lines Col-0, *drb1*, *drb2* and *drb4* post exposure to a 7-day period of PO_4_ starvation. (**A**) The aerial tissue phenotypes expressed by non-stressed (top row of panels) and PO_4_-stressed (bottom row of panels) 15-day old Col-0, *drb1*, *drb2* and *drb4* plants. Scale bar = 1cm. (**B**) Quantification of the shoot mass of 8-day old Col-0, *drb1*, *drb2* and *drb4* seedlings germinated and cultivated under standard growth conditions. (**C**) The shoot mass of non-stressed and PO_4_-stressed 15-day old Col-0, *drb1*, *drb2* and *drb4* plants. (**D**) The rosette area of non-stressed and PO_4_-stressed 15-day old *Arabidopsis* lines, Col-0, *drb1*, *drb2* and *drb4*. (**E**) Anthocyanin accumulation in the shoot tissues of 15-day old Col-0, *drb1*, *drb2* and *drb4* plants cultivated under standard growth conditions, or for 7-days under PO_4_ starvation. (**F** and **G**) Chlorophyll *a* (**F**) and chlorophyll *b* (**G**) abundance in the aerial tissues of non-stressed and PO_4_-stressed Col-0, *drb1*, *drb2* and *drb4* plants. (**B**-**G**) Error bars represent the standard deviation of four biological replicates and each biological replicate consisted of a pool of twelve individual plants. The presence of an asterisk above a column represents a statistically significant difference either between non-stressed Col-0 plants and each assessed *drb* mutant post cultivation under either a non-stressed or stressed growth regime (**B**) or between the non-stressed and PO_4_-stressed sample of each plant line (**C**-**G**) (*p*-value: * < 0.05; ** < 0.005; *** < 0.001).

**Figure 2 plants-08-00124-f002:**
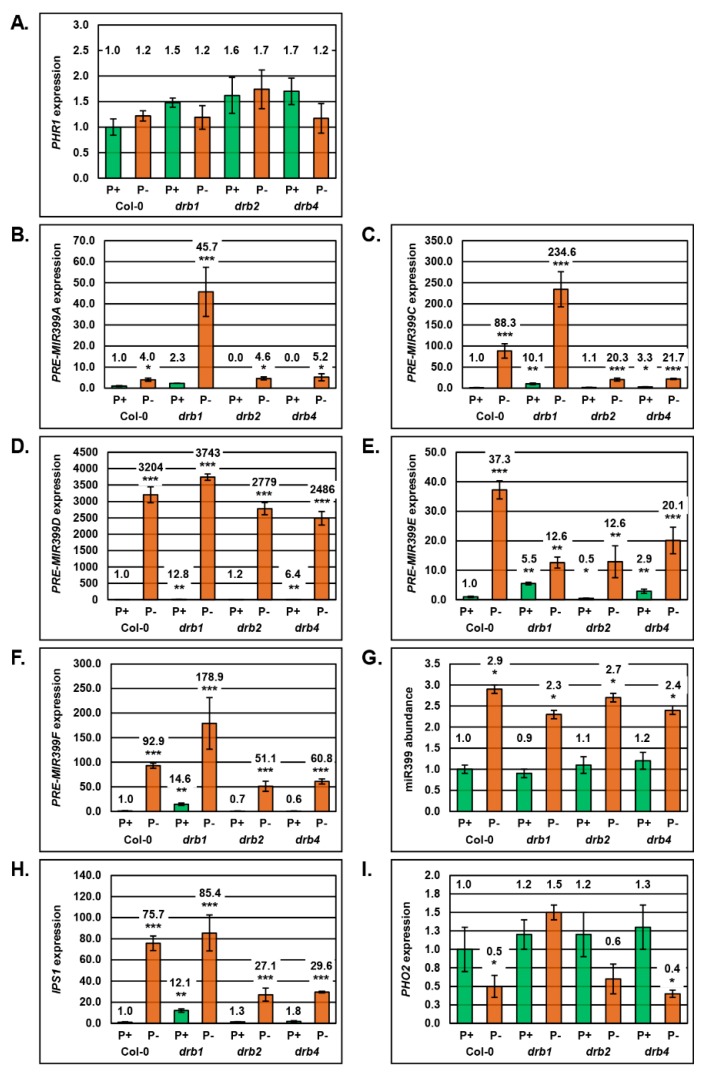
Molecular profiling of the miR399/*PHO2* expression module in the aerial tissues of non-stressed and PO_4_-stressed Col-0, *drb1*, *drb2* and *drb4* plants. (**A**) RT-qPCR assessment of the expression of the PO_4_ responsive transcription factor PHR1 in the aerial tissues of non-stressed and PO_4_-stressed Col-0, *drb1*, *drb2* and *drb4* plants. (**B** to **F**) RT-qPCR profiling of miR399 precursor transcript abundance in the aerial tissues of non-stressed and PO_4_-stressed Col-0, *drb1*, *drb2* and *drb4* plants, including precursors *PRE-MIR399A* (**B**), *PRE-MIR399C* (**C**), *PRE-MIR399D* (**D**), *PRE-MIR399E* (**E**) and *PRE-MIR399F* (**F**). (**G**) Quantification of miR399 abundance in the aerial tissues of non-stressed and PO_4_-stressed Col-0, *drb1*, *drb2* and *drb4* plants. (**H**) Assessment of the expression of the non-cleavable decoy of miR399 activity, *IPS1*, via RT-qPCR in the aerial tissues of non-stressed and PO_4_-stressed *Arabidopsis* lines, Col-0, *drb1*, *drb2* and *drb4*. (**I**) RT-qPCR analysis of the expression of the miR399 target gene, *PHO2*, in the aerial tissues of non-stressed and PO_4_-stressed *Arabidopsis* lines, Col-0, *drb1*, *drb2* and *drb4*. (**A–I**) Error bars represent the standard deviation of four biological replicates and each biological replicate consisted of a pool of twelve individual plants. Due to the vastly different levels of each assessed transcript, the relative expression value for each plant line/growth regime is provided above the corresponding column. The presence of an asterisk above a column represents a statistically significant difference between non-stressed Col-0 plants and each of the assessed *drb* mutant lines, post cultivation under either a standard or stressed growth regime (*p*-value: * < 0.05; ** < 0.005; *** < 0.001).

**Figure 3 plants-08-00124-f003:**
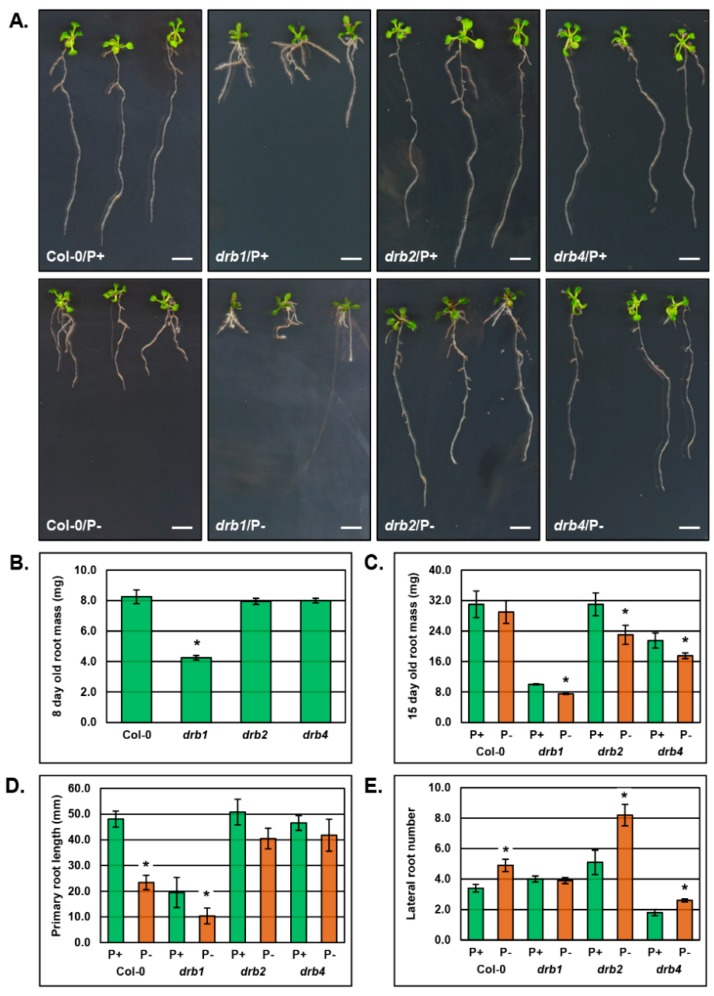
The root system phenotypes displayed by 15-day old *Arabidopsis* plant lines Col-0, *drb1*, *drb2* and *drb4* post exposure to a 7-day period of PO_4_ starvation. (**A**) The root system phenotypes expressed by non-stressed (top row of panels) and PO_4_-stressed (bottom row of panels) 15-day old Col-0, *drb1*, *drb2* and *drb4* plants. Scale bar = 1cm. (**B**) Quantification of the root mass of 8-day old Col-0, *drb1*, *drb2* and *drb4* seedlings cultivated under standard growth conditions. (**C**) The root mass of non-stressed and PO_4_-stressed 15-day old Col-0, *drb1*, *drb2* and *drb4* plants. (**D**) The primary root length of non-stressed and PO_4_-stressed 15-day old *Arabidopsis* lines, Col-0, *drb1*, *drb2* and *drb4*. (**E**) The number of lateral roots formed from the primary root of 15-day old Col-0, *drb1*, *drb2* and *drb4* plants cultivated under standard growth conditions, or post the 7-day PO_4_ starvation period. (**B–E**) Error bars represent the standard deviation of four biological replicates and each biological replicate consisted of a pool of twelve individual plants. The presence of an asterisk above a column represents a statistically significant difference either between non-stressed Col-0 plants and each assessed *drb* mutant post cultivation under either a non-stressed or stressed growth regime (**B**) or between the non-stressed and PO_4_-stressed sample of each plant line (**C-E**) (*p*-value: * < 0.05; ** < 0.005; *** < 0.001).

**Figure 4 plants-08-00124-f004:**
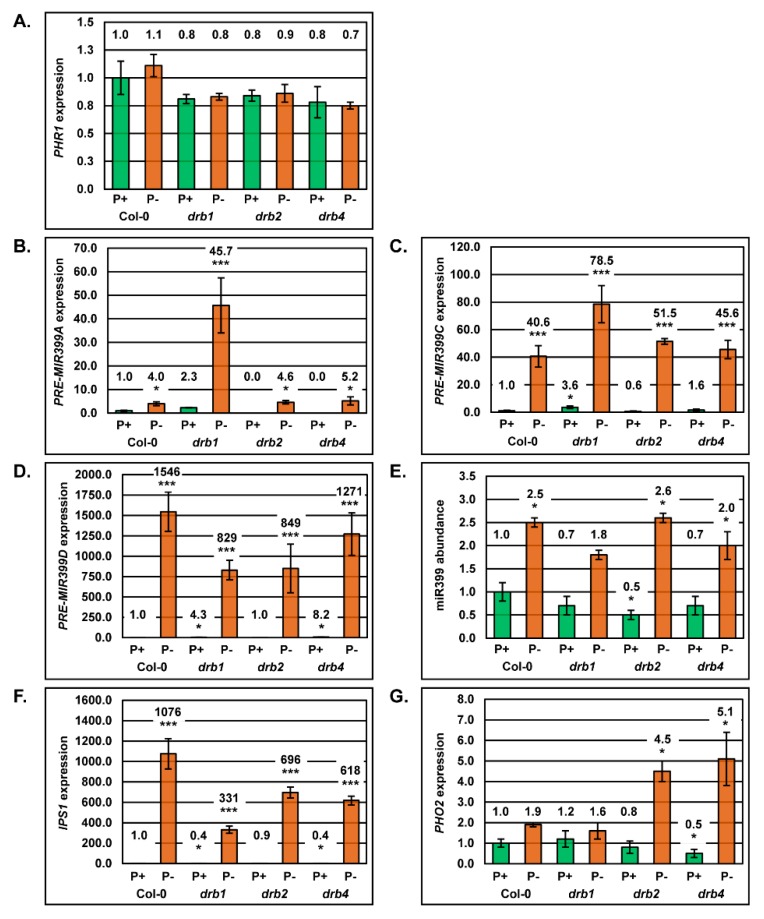
Molecular profiling of the miR399/*PHO2* expression module in the root system of non-stressed and PO_4_-stressed Col-0, *drb1*, *drb2* and *drb4* plants. (**A**) RT-qPCR assessment of the expression of the PO_4_ responsive transcription factor *PHR1* in the roots of non-stressed and PO_4_-stressed Col-0, *drb1*, *drb2* and *drb4* plants. (**B–D**) RT-qPCR profiling of miR399 precursor transcript abundance in the root system of non-stressed and PO_4_-stressed Col-0, *drb1*, *drb2* and *drb4* plants, including precursors *PRE-MIR399A* (**B**), *PRE-MIR399C* (**C**) and *PRE-MIR399D* (**D**). (**E**) Quantification of miR399 abundance in the roots of non-stressed and PO_4_-stressed Col-0, *drb1*, *drb2* and *drb4* plants. (**F**) Assessment of *IPS1* transcript abundance in the roots of non-stressed and PO_4_-stressed *Arabidopsis* lines, Col-0, *drb1*, *drb2* and *drb4*. (**G**) RT-qPCR analysis of *PHO2* expression, the target gene of miR399, in the root system of non-stressed and PO_4_-stressed *Arabidopsis* lines, Col-0, *drb1*, *drb2* and *drb4*. (**A–G**) Error bars represent the standard deviation of four biological replicates and each biological replicate consisted of a pool of twelve individual plants. Due to the vastly different level of each assessed transcript, the relative expression value for each plant line/growth regime is provided above the corresponding column. The presence of an asterisk above a column represents a statistically significant difference between non-stressed Col-0 plants and each of the assessed *drb* mutant lines, post cultivation under either a standard or stressed growth regime (*p*-value: * < 0.05; ** < 0.005; *** < 0.001).

**Figure 5 plants-08-00124-f005:**
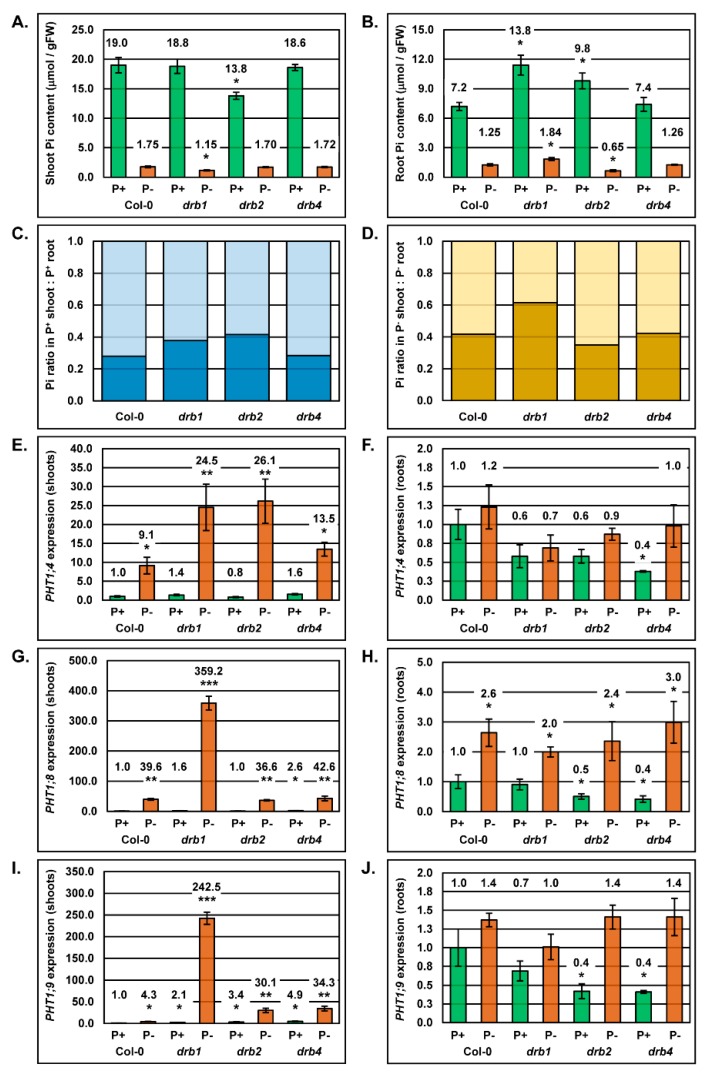
Pi content and PO_4_ transporter gene expression in the shoot and root tissue of 15-day old *Arabidopsis* plant lines Col-0, *drb1*, *drb2* and *drb4* cultivated under either a standard growth regime or post-exposure to a 7-day period of PO_4_ starvation. (**A,B**) Comparison of the Pi content of the shoots (A) and roots (B) of 15-day old non-stressed and PO_4_-stressed Col-0, *drb1*, *drb2* and *drb4* plants. (**C**) Pi content shoot (light blue) to root (dark blue) ratio of 15-day old Col-0, *drb1*, *drb2* and *drb4* plants cultivated under standard growth conditions. (**D**) Pi content shoot (light gold) to root (dark gold) ratio of 15-day old Col-0, *drb1*, *drb2* and *drb4* plants post 7-days of PO_4_ starvation. (**E,F**) Quantification of *PHT1;4* expression in the shoot (E) and root (F) tissues of 15-day old Col-0, *drb1*, *drb2* and *drb4* plants cultivated under standard growth conditions or post a 7-day period of PO_4_ starvation. (**G,H**) RT-qPCR assessment of *PHT1;8* transcript abundance in the shoots (G) and roots (H) of 15-day old Col-0, *drb1*, *drb2* and *drb4* plants cultivated under either standard or PO_4_ stress conditions. (**I,J**) *PHT1;9* expression in the shoot (I) and root (J) material of non-stressed or PO_4_-stressed Col-0, *drb1*, *drb2* and *drb4* plants at 15 days of age. (A,B,E–J) Error bars represent the standard deviation of four biological replicates and each biological replicate consisted of a pool of twelve individual plants. Due to the vastly different levels of each assessed transcript, the relative expression value for each plant line/growth regime is provided above the corresponding column. The presence of an asterisk above a column represents a statistically significant difference between the non-stressed and PO_4_-stressed sample of each plant line (A,B) or between non-stressed Col-0 plants and each *drb* mutant line, post cultivation under either a standard or stressed growth regime (E–J) (*p*-value: * < 0.05; ** < 0.005; *** < 0.001).

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
