# Peer review of "DRB1, DRB2 and DRB4 Are Required for Appropriate Regulation of the microRNA399/PHOSPHATE2 Expression Module in Arabidopsis thaliana"

_plants, 2019, doi:10.3390/plants8050124_

Reviewer 1 Report

Review

In the work entitled “DRB1, DRB2 and DRB4 are required for appropriate regulation of the microRNA399/ PHOSPHATE2 expression module in Arabidopsis thaliana” the authors analyzed the role of DRB1, DRB2 and DRB4 in the regulation of the miR399/PHO2 expression module when plants are growing under PO4 starvations conditions.

More and more the study of abiotic stress growth conditions for plants is necessary in order to find solutions for proper crop’s growth especially in places where soils are poor in nutrients. Therefore, this work is important and brings new insights about genetic regulation occurring in plants when facing the lack of available phosphate in the soil.

The experiments are well designed, the manuscript is well organized and the pictures and graphs are of good quality.

I do have a major concern regarding the way it is written, though. I reckon the person who wrote the manuscript took great pleasure in doing it taking advantage of all she/he writing skills but for the rest of us, mundane scientists, we crave for direct, short and straight to the point papers. With that being said, the manuscript needs to be reduced in 60% (this is not an over statement). It´s very long and very dense making the read of it extenuating. I am giving some examples but the all text needs major review regarding the size

Ex:

In the results section, the authors discuss too much. Must of it should be moved to the discussion section that in part, is already there.

Page 6, lines 254 – 258 and lines 261-266. These sentences are too long.

Page 7, lines 328-334 – Please, remove all this. It doesn´t make any sense all this information in the results.

Description of Fig. 2 starts on page 5 (top) and it ends middle page 8. It´s too extensive.

Page 8, lines 374 – 382. Please remove all this or adjust it in the discussion. It´s too noisy.

Page 8, lines 382-384. Please, reduce these 3 lines sentence to just one.

Page 8, lines 384 – 390. Again, all this is perfectly avoidable. Just describe the results in a simple and direct fashion.

Page 8, lines 400 – 402. Please, just remove this line all together. It doesn´t add anything.

The authors use words such as “surprisingly” and “surprising” too often.

It´s not necessary to present again, data in the discussion. Ex. Page 14, lines 702-705. Lines 775 – 784 are pure results and already present in the results.

Another example in material and methods section; page 21, lines 1023-1026. Completely unnecessary.

Major review

-          Page 4, line 155 – The authors refer to the 7 day PO4 stress treatment (please add a fig. number right after the describing text) and then, they move directly to the results of 15 days old plants. Please, be more precise in the description always indicating what figure the authors are referring to.

              Minor review

-          Page 6, line 259. PRE-MIR399E instead of PRE-MIE399E.

Author Response

Dear Reviewer #1;

Thank you kindly for taking the time to review our submitted manuscript. Your efforts and helpful suggestions are very much appreciated by the authorship team. We have made considerable effort to shorten the length of the original submission with the revised submission reduced in length by 5 pages of text (clean copy (post removal of tracked changes) is reduced by 5 pages / the tracked changes copy requested by the journal is increased in page length due to number of changes that we have made in the revised submission). We are concerned however, that further removal of additional text will result in loss of the narrative of the findings that we are presenting in this revised submission. Below we supply a point-by-point response to your major concerns with the original submission, concerns that we hope that you find we have addressed to a satisfactory level in the revised version of our manuscript.

1.   I do have a major concern regarding the way it is written, though. I reckon the person who wrote the manuscript took great pleasure in doing it taking advantage of all she/he writing skills but for the rest of us, mundane scientists, we crave for direct, short and straight to the point papers. With that being said, the manuscript needs to be reduced in 60% (this is not an over statement). It´s very long and very dense making the read of it extenuating. I am giving some examples but the all text needs major review regarding the size.

-       We have greatly reduced the volume of the text in the revised version of our manuscript. Specifically, 5 pages of textual description of our presented results have been removed from the revised version of the manuscript. As you have suggested below, a lot of the text that we have either removed, or summarised, is from the results section of the manuscript, including those examples that you kindly provided post your original assessment of our study.

2.   In the results section, the authors discuss too much. Must of it should be moved to the discussion section that in part, is already there.

-       We have removed a lot of what was originally contained in the results section of our original submission in the revised submission. The revised submission is reduced in length by 5 pages of text. This reduction was primarily achieved via summarisation of the presented results.

3.   Page 6, lines 254 – 258 and lines 261-266. These sentences are too long.

-       Thank you for pointing these specific sentences out. They have been addressed in the revised submission

4.   Page 7, lines 328-334 – Please, remove all this. It doesn´t make any sense all this information in the results.

-       As suggested post your review of our original submission, we have removed this section of text from the revised submission.

5.    Description of Fig. 2 starts on page 5 (top) and it ends middle page 8. It´s too extensive.

-       Thank you again for raising this concern on the length of description of a single Figure. We have extensively summarised our description of the results presented in this Figure in the revised submission.

6.   Page 8, lines 374 – 382. Please remove all this or adjust it in the discussion. It´s too noisy.

-       As suggested, we have removed this section of text in the revised version of the manuscript.

7.   Page 8, lines 382-384. Please, reduce these 3 lines sentence to just one.

-       This suggestion has been addressed in the revised version of the manuscript.

8.   Page 8, lines 384 – 390. Again, all this is perfectly avoidable. Just describe the results in a simple and direct fashion.

-       Thank you for pointing out this issue. We have addressed this concern in the revised manuscript.

9.   Page 8, lines 400 – 402. Please, just remove this line all together. It doesn´t add anything.

-       Thank you again for pointing out this specific issue. We have removed this from the revised version of the manuscript.

10.    The authors use words such as “surprisingly” and “surprising” too often.

-       Thank you for identifying this issue. We have removed many numerous uses of both words.

11.    It´s not necessary to present again, data in the discussion. Ex. Page 14, lines 702-705. Lines 775 – 784 are pure results and already present in the results.

-       The related descriptions in the Results and Discussions section of the revised manuscript are now distinct to one another.

12.    Another example in material and methods section; page 21, lines 1023-1026. Completely unnecessary.

-       This issue has been addressed in the revised manuscript.

Major review

13.    Page 4, line 155 – The authors refer to the 7 day PO4stress treatment (please add a fig. number right after the describing text) and then, they move directly to the results of 15 days old plants. Please, be more precise in the description always indicating what figure the authors are referring to.

-       Thank you again for identifying this oversight. We have added additional Figure references through the text of the revised manuscript.

Minor review

14.    Page 6, line 259. PRE-MIR399E instead of PRE-MIE399E.

Thank you for identifying this error. Much appreciated, this corrected in the revised version the manuscript.

Reviewer 2 Report

DRB1, DRB2 and DRB4 are required for appropriate regulation of the microRNA399/ PHOSPHATE2 expression module in Arabidopsis thaliana

In this manuscript Pegler et al., attempted to establish the importance of DRB1, DRB2 and DRB4 in P homeostasis in A. thaliana through miR399/PHO2 pathway. Authors quantified the transcript levels of major players of the pathway in drb1, drb2 and drb4 mutant plants to that of the col-0 wild type via qRT-PCR under P-sufficient and P-deficient conditions to establish the importance (Figures 2 and 4). Some phenotyping has already been done for DRB mutants in A. thaliana. Authors have redone some of them while adding few more (Figures 1 and 3). Changes in Pi levels and transcript levels of phosphate transporters PHT1;4, PHT1;8 and PHT1;9 in shoots and roots of drb1, drb2 and drb4 mutant plants show a good correlation at least in the shoots (Figure 5) providing considerable strength to the manuscript. The findings are of interest to the Arabidopsis/plant research community in general and researchers working on the molecular mechanisms of P homeostasis in particular.

However, I have the following  2 major concerns:

Major Concerns

1.     Although the transcript levels of miR399 precursors, miR399 itself, IPS1and PHO2 in shoot and roots are altered in the drb1, drb2 and drb4 mutants to various levels compared to col-0 wild type (Figures 2 and 4) they do not correlate very well with the proposed model especially in roots (Additionally PHO2 levels in col-0 in P-starvation contradicts the proposed model in Figure 4). The data presented suggest potential functional redundancy among the DRBs tested and therefore the level of functional redundancy should be assessed. Moreover, the authors claim that DRB1 plays a main role in miR399 production and DRB2/DRB4 plays a secondary role most probably an antagonistic one in shoots (Figure 2 and published data). Although, these claims may be sought from the data, it is necessary to assess the miR399 precursor, miR399 and PHO2 levels in higher order mutants (double mutants drb1 drb2, drb1 drb3, drb1 drb2 drb4 triple mutants etc.) to undoubtedly arrive at such conclusions and to rule out involvement of other players reported to be involved in P homeostasis.

2.     Describing statistically non-significant subtle changes in length, making conclusions based on such changes is prominent throughout the manuscript and also making claims or conclusions from non-significant subtle changes only whenever those fits in to authors’ hypothesis. Since these may be mere biological noise, avoid lengthy reporting and discussion of such changes. Please find some examples below.

Examples:

–    Fig. 2A: Are PHR1 levels statistically significant? If not please moderate the section

      between Lines 218 – 230. Same is true for Fig 4A

      Fig 2I: PHO2 levels: claims made Lines 354 – 364

      Fig 3D: drb2 and drb4 primary root length

General Comments

1.     Resolution of figures are poor including axes titles in graphs especially Fig 1.

Authors assessed the transcript levels 7 days after transferring to P-deficient medium. This may be appropriate to assess physiological changes. Although, miR399 and IPS1 levels are still higher 7 days after transferring to the P-deficient medium, PHR1 and PHO1 mRNA levels may have already reached an equilibrium. Therefore, it may be useful to use few time points including an early time point to see whether there are detectable changes in PHO2 mRNA levels in the mutants compared to col-0 wild type.  

Author Response

Dear Reviewer #2,

The authors thank you for taking the time to critically review our submitted study. We have addressed the numerous concerns raised by Reviewer #1 that primary regarded the length of the original submission. Namely, the length of the manuscript has been reduced by 5 pages of text. This was achieved by summarising the results in a more concise and focused manner (a similar comment to what you have made). We hope that you find the changes that we have made have improved the quality of the revised manuscript. Below, we provide a point-by-point summary addressing the specific concerns that you raised following your review of our original submission. 

1.   Although the transcript levels of miR399 precursors, miR399itself, IPS1and PHO2in shoot and roots are altered in the drb1drb2and drb4mutants to various levels compared to col-0 wild type (Figures 2 and 4) they do not correlate very well with the proposed model especially in roots (Additionally PHO2levels in col-0 in P-starvation contradicts the proposed model in Figure 4). 

-       We have addressed this concern via provision of an alternate model in an attempt to explain the complex molecular profiles that we were presented with in this study. We thank you for pointing out this issue that we neglected in our original submission. Below is the additional text that we have included on page 12, lines 2743 to 2752 of the revised manuscript, in an attempt to address your concern.

“Alternatively, elevated PHO2expression in P+Col-0 and P+drb1roots when miR399 abundance is also elevated may result from the enhanced expression of the eTM of miR399 activity, IPS1. In P-Col-0 shoots for example, where elevated miR399 abundance was demonstrated to direct enhanced expression repression of the PHO2transcript (Figure 2G,2I), IPS1abundance was elevated by 75.7-fold, compared to its abundance in P+Col-0 shoots (Figure 2H). In PO4-stressed roots however, IPS1expression was elevated to a much greater degree, by 1076-fold (Figure 4F). This 14.2-fold greater promotion to IPS1expression in P-Col-0 roots, than that observed in P-Col-0 shoots, would be expected to sequester a higher amount of miR399, which in turn, could have led to the observed elevation in PHO2expression in P-Col-0 roots in the presence of 2.5-fold greater abundance of the PHO2targeting miRNA, miR399.”

2.   The data presented suggest potential functional redundancy among the DRBs tested and therefore the level of functional redundancy should be assessed. Moreover, the authors claim that DRB1 plays a main role in miR399 production and DRB2/DRB4 plays a secondary role most probably an antagonistic one in shoots (Figure 2 and published data).

-         We have previously demonstrated highly similar trends in precursor transcript abundance and miRNA accumulation for a range of miRNAs in the drb2mutant background that indicate antagonism of DRB1 function by DRB2 in the production stage of the miRNA pathway (Eamens et al. 2012, PLoS One and Eamens et al., 2012, Plant Signal Behav). DRB2 has also been demonstrated to be antagonistic to the function of DRB4 in the DRB4/DCL4 partnership (Pélissier et al., 2011) for both miRNA and siRNA production. We therefore are of the opinion that we have presented enough additional evidence here to make the statement that DRB1 is the primary DRB required for miR399 production from its precursor transcripts and that both DRB2 and DRB4 play secondary roles in this process in Arabidopsis shoot and root tissue.

3.   Although, these claims may be sought from the data, it is necessary to assess the miR399 precursor, miR399 and PHO2levels in higher order mutants (double mutants drb1 drb2,drb1 drb3drb1 drb2 drb4triple mutants etc.) to undoubtedly arrive at such conclusions and to rule out involvement of other players reported to be involved in P homeostasis.

-         Yes, we totally agree your suggestion of assessment of higher order mutants for more confident assignment of functional redundancy. Unfortunately however, since relocating my research group to my current institute, we no longer have access to the higher order drb mutant combinations that we have published on previously (Curtin et al., 2008, FEBS Letts; Eamens et al., 2012, PLoS One; Reis et al., 2015, Nature Plants) that would be required to facilitate this additional research. Furthermore, the plant growth facility used to perform the analyses reported on here in this study has been demolished and we are still waiting on our new facility to (1) be built and then (2) come online for research use. Therefore, we cannot perform any additional experimentation at this time sorry.

4.   Describing statistically non-significant subtle changes in length, making conclusions based on such changes is prominent throughout the manuscript and also making claims or conclusions from non-significant subtle changes only whenever those fits in to authors’ hypothesis. Since these may be mere biological noise, avoid lengthy reporting and discussion of such changes. Please find some examples below.

-       A similar comment was made by Reviewer #1 post their review of our original submission. In the revised version of our manuscript, we have made considerable efforts to address this concern. Specifically, we have reduced the length of the original manuscript by five pages of text. A lot of the original text has been summarised in the revised submission, and as part of this process, we have tried to avoid the reporting of subtle non-significant changes and making conclusions based on such data.

5.  Examples: (1) Fig. 2A: Are PHR1levels statistically significant? If not please moderate the section between Lines 218 – 230. (2) Same is true for Fig 4A(2) Fig 2I: PHO2levels: claims made Lines 354 – 364 (4) Fig 3D: drb2and drb4primary root length.

-       Thank you for identifying your specific concerns. Please see point #4 above. We have reduced the length of the revised manuscript by 5 pages. This was primarily achieved via summarising the results presented in manuscript Figures 1 to 5 in a more precise and focused manner. As part of this process we believe that we have addressed each of the specific concerns that you raised above as part of your review of the original submission. 

6.  Resolution of figures are poor including axes titles in graphs especially Fig 1.

-        Thank you for identifying this oversight. The resolution of each of the five Figures presented in the revised manuscript has been reformatted in order to improve the resolution and thus overall quality of each Figure.

7.  Authors assessed the transcript levels 7 days after transferring to P-deficient medium. This may be appropriate to assess physiological changes. Although, miR399and IPS1levels are still higher 7 days after transferring to the P-deficient medium, PHR1and PHO1mRNA levels may have already reached an equilibrium. Therefore, it may be useful to use few time points including an early time point to see whether there are detectable changes in PHO2mRNA levels in the mutants compared to col-0 wild type.

-       Yes, we totally agree with your informative suggestion that assessment of the expression of a higher order, primary transcription factor such as PHR1, is likely to occur at an earlier time point than the 7 day time point that we used for sampling in this study. Unfortunately, as stated in point #3 above, due to the plant growth facility that the reported on experimentation was conducted in having been demolished prior to the construction of our new plant growth facility being completed, we currently to do have access to an appropriate growth facility in order to conduct any additional experimentation. However, the low level of expression induction observed for PHR1 post the 7 day PO4 starvation period imposed on the four Arabidopsis lines assessed in this study closely matches the low level of PHR1 expression induction reported previously by Rubio et al., (2001). 

Round  2

Reviewer 1 Report

In the work entitled “DRB1, DRB2 and DRB4 are required for appropriate regulation of the microRNA399/ PHOSPHATE2 expression module in Arabidopsis thaliana” the authors analyzed the role of DRB1, DRB2 and DRB4 in the regulation of the miR399/PHO2 expression module when plants are growing under PO4 starvations conditions.

More and more the study of abiotic stress growth conditions for plants is necessary in order to find solutions for proper crop’s growth especially in places where soils are poor in nutrients. Therefore, this work is important and brings new insights about genetic regulation occurring in plants when facing the lack of available phosphate in the soil.

Thank you for addressing all the comments. Again this is an important work and I have no further reviews.

Reviewer 2 Report

Satisfied with the responses provided by the authors to my initial concerns/ comments.